# Polyploidy in the adult *Drosophila* brain

**Shyama Nandakumar, Olga Grushko, Laura A Buttitta\***

Molecular, Cellular, and Developmental Biology, University of Michigan, Ann Arbor, United States

**Abstract** Long-lived cells such as terminally differentiated postmitotic neurons and glia must cope with the accumulation of damage over the course of an animal's lifespan. How long-lived cells deal with ageing-related damage is poorly understood. Here we show that polyploid cells accumulate in the adult fly brain and that polyploidy protects against DNA damage-induced cell death. Multiple types of neurons and glia that are diploid at eclosion, become polyploid in the adult *Drosophila* brain. The optic lobes exhibit the highest levels of polyploidy, associated with an elevated DNA damage response in this brain region. Inducing oxidative stress or exogenous DNA damage leads to an earlier onset of polyploidy, and polyploid cells in the adult brain are more resistant to DNA damage-induced cell death than diploid cells. Our results suggest polyploidy may serve a protective role for neurons and glia in adult *Drosophila melanogaster* brains.

## Introduction

Terminally differentiated postmitotic cells such as mature neurons and glia are long-lived and must cope with the accumulation of damage over the course of an animal's lifespan. The mechanisms used by such long-lived cells to deal with aging-related damage are poorly understood. The brain of the fruit fly *Drosophila melanogaster* is an ideal context to examine this since the fly has a relatively short lifespan and the adult fly brain is nearly entirely postmitotic with well understood development and excellent tools for genetic manipulations.

The adult central nervous system of *Drosophila melanogaster* comprises ~110,000 cells, most of which are generated in the larval and early pupal stages of development from various progenitor cell types (*Truman and Bate, 1988*; *White and Kankel, 1978*). By late metamorphosis, the *Drosophila* pupal brain is normally completely non-cycling and negative for markers of proliferation such as thymidine analog incorporation and mitotic markers (*Awasaki et al., 2008*; *Pahl et al., 2019*; *Siegrist et al., 2010*).

In the adult, very little neurogenesis and gliogenesis are normally observed (*Awasaki et al., 2008*; *Ito and Hotta, 1992*; *von Trotha et al., 2009*). A population of about 40 adult neural progenitors has been reported in the optic lobe and a population of glial progenitors has been reported in the central brain (*Fernández-Hernández et al., 2013*; *Foo et al., 2017*). Upon damage or cell loss, hallmarks of cycling have been shown to be activated, although the overall level of proliferation in the adult brain remains very low (*Crocker et al., 2020*; *Fernandez-Hernandez et al., 2019*; *Fernández-Hernández et al., 2013*; *Foo et al., 2017*; *Li et al., 2020*). Thus, the brain of the adult fly is thought to be almost entirely postmitotic with most cells in G0 with a diploid (2C) DNA content. One known exception to this are the cells that constitute the 'blood-brain barrier' of *Drosophila*.

The 'blood-brain barrier' in *Drosophila* is made up of specialised cells called the Sub-perineurial glia (SPGs). These cells are very few in number and achieve growth without cell division by employing variant cell cycles termed endocycles, that involve DNA replication without karyokinesis or cytokinesis, as well as endomitotic cycles that involve DNA replication and karyokinesis without cytokinesis (*Unhavaithaya and Orr-Weaver, 2012*; *Von Stetina et al., 2018*). The SPGs undergo these variant cell cycles to increase their size rapidly to sustain the growth of the underlying brain during larval development. The polyploidization of these cells plays an important role in maintaining

**\*For correspondence:**
buttitta@umich.edu

**Competing interests:** The authors declare that no competing interests exist.

their epithelial barrier function, although it remains unclear whether these cells continue to endocycle or endomitose in the adult.

Polyploidy can also confer an increased biosynthetic capacity to cells and resistance to DNA damage induced cell death (*Edgar and Orr-Weaver, 2001*; *Lee et al., 2009*; *Mehrotra et al., 2008*; *Zhang et al., 2014*). Several studies have noted neurons and glia in the adult fly brain with large nuclei (*Robinow and White, 1991*; *Winberg et al., 1992*) and in some cases neurons and glia of other insect species in the adult CNS are known to be polyploid (*Nordlander and Edwards, 1969*). Rare instances of neuronal polyploidy have been reported in vertebrates under normal conditions (*Morillo et al., 2010*) and even in the CNS of mammals (*López-Sánchez and Frade, 2013*; *Shai et al., 2015*).

Polyploidization is employed in response to tissue damage and helps maintain organ size (*Cohen et al., 2018*; *Tamori and Deng, 2013*; *Losick et al., 2016*; *Losick et al., 2013*). Therefore, polyploidy may be a strategy to deal with damage accumulated with age in the brain, a tissue with very limited cell division potential. Here we show that polyploid cells accumulate in the adult fly brain and that this proportion of polyploidy increases as the animals approach middle-age. We show that multiple types of neurons and glia which are diploid at eclosion which become polyploid specifically in the adult brain. We have found that the optic lobes of the brain contribute to most of the observed polyploidy. We also observe increased DNA damage with age, and show that inducing oxidative stress and exogenous DNA damage can lead to increased levels of polyploidy. We find that polyploid cells in the adult brain are resistant to DNA damage-induced cell death and propose a potentially protective role for polyploidy in neurons and glia in adult *Drosophila melanogaster* brains.

## Results

Cell ploidy often scales with cell size and biosynthetic capacity (*Edgar and Orr-Weaver, 2001*; *Orr-Weaver, 2015*). The brain is thought to be a notable exception to this rule, where the size of postmitotic diploid neurons and glia can be highly variable. We wondered whether alterations in ploidy during late development or early adulthood may contribute to the variability in neuronal and glial cell size in the mature *Drosophila* brain. We therefore developed a sensitive flow cytometry assay to measure DNA content in *Drosophila* pupal and adult brains. Briefly, this assay involves dissociating brains with a trypsin or collagenase based solution followed by quenching the dissociation and labeling DNA with DyeCycle Violet (*Grushko and Buttitta, 2015*) in the same tube, to avoid cell loss from washes. Samples are then immediately run live on a flow cytometer for analysis. We employed strict gating parameters to eliminate doublets (*Figure 1—figure supplement 1*; *López-Sánchez et al., 2017b*). This assay is sensitive enough to measure DNA content from small subsets of cells (e.g. *Mz19-GFP* expressing neurons) from individual pupal or adult brains (*Figure 1—figure supplement 1D*). Using this approach we confirmed that under normal culturing conditions, cell proliferation and DNA replication ceases in the pupal brain after 24 hr into metamorphosis (24 hr APF) (*Figure 1—figure supplement 1E*), and that only the previously described mushroom body neuroblasts continue to replicate their DNA and divide in late pupa (*Siegrist et al., 2010*). The brains of newly eclosed adult flies are 98–99% diploid and we, like others, only rarely observe EdU incorporation during the first week of adulthood in wild-type flies under normal conditions but not later in adulthood (*Figure 1—figure supplement 1F*; *Fernández-Hernández et al., 2013*; *Foo et al., 2017*; *Kato et al., 2009*; *Siegrist et al., 2010*; *von Trotha et al., 2009*). We were therefore surprised to find a distinct population of cells with DNA content of 4C and up to >16C appearing in brains of aged animals of various genotypes under normal culture conditions (*Figure 1—figure supplement 1G*).

### Polyploid cells accumulate in the adult *Drosophila* brain

We performed a systematic time-course to measure accumulation of polyploid cells in the adult brain in isogenic $w^{1118}$ male and female flies cultured under standard conditions (*Linford et al., 2013*). We measured the percentage of polyploid cells in individual brains from the day of eclosion until 56 days (8 weeks) at weekly intervals. Polyploid cells appear as early as 7 days into adulthood, and the proportion of polyploidy continues to rise until animals are 21 days old (*Figure 1A*). This increase in polyploidy is only observed until week 3, after which the proportion of polyploid cells observed

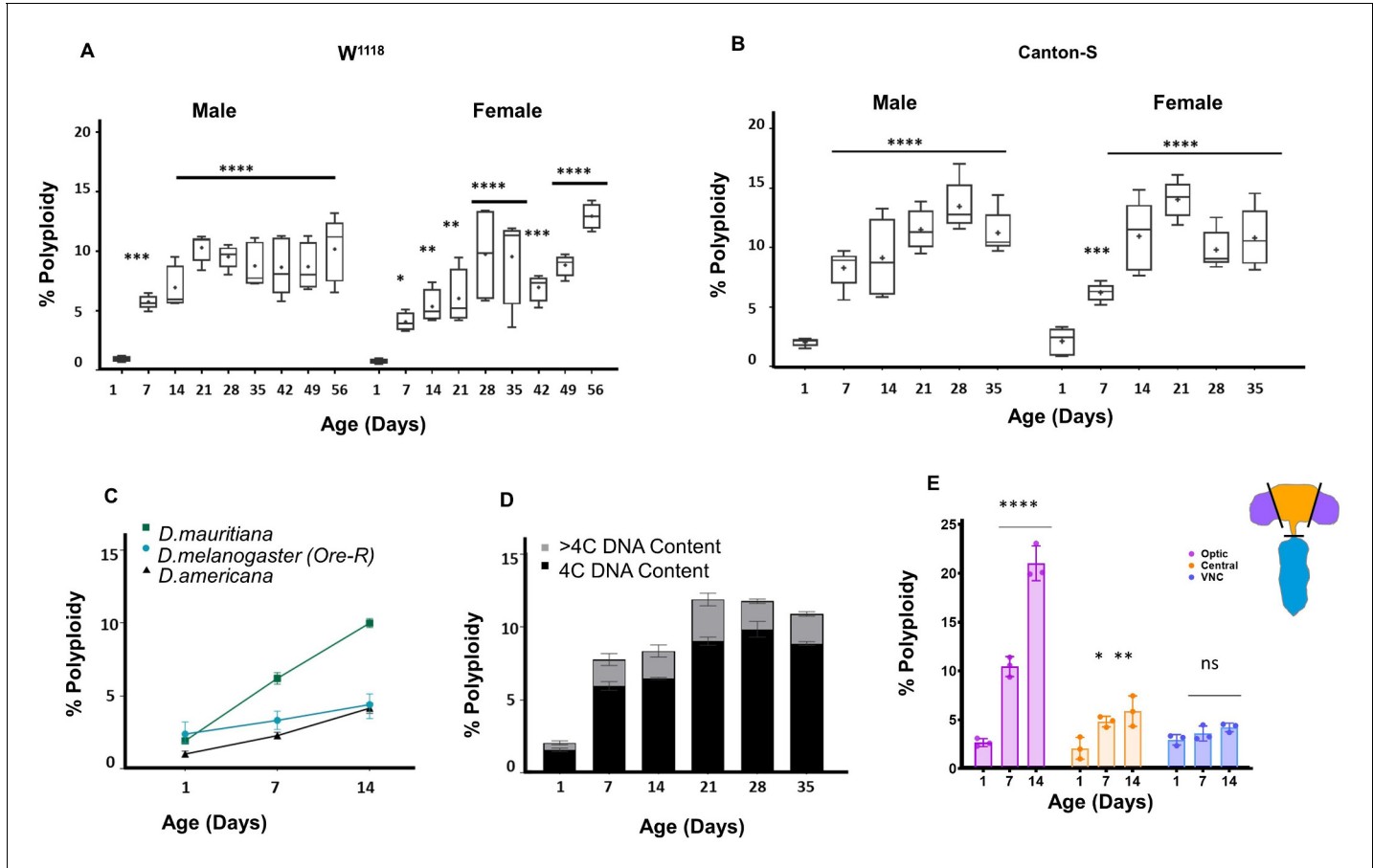

**Figure 1.** Polyploid cells accumulate in the adult *Drosophila* brain. (A,B) Percentage of cells in individual brains exhibiting polyploidy in *w1118* (A) and *Canton-S* (B) male and female whole brains. Age in days indicates days post-eclosion. Box plots showing range, dot indicates mean (n = 10). (Two-way ANOVA with greenhouse-geisser correction for unequal SDs followed by Holm-Sidak's multiple comparisons test. P values: ns > 0.1234;<0.0332 *; <0.0021 **;<0.0002 ***; ****<0.0001) (C) Accumulation of polyploidy is also observed in other *Drosophila* species. *D.mauritiana* and *D.americana* shown respectively in green and black compared to *Oregon-R* (*D. melanogaster)* shown in teal at different time points post-eclosion. Shapes indicate mean polyploidy observed, bars show range. three brains each per sample, n = 2 per time point. (D) Stacked bar plot showing proportion of polyploid cells with tetraploid or 4C DNA content (black) and greater than tetraploid or >4C DNA content (grey) in *Canton-S* males at different ages. (D) Percentage of polyploidy in Optic lobes (OL shown in purple), central brain (CB, shown in orange) and ventral nerve cord (VNC shown in blue) at different ages, *w1118* (Error bars show mean ± SEM n = 3).

The online version of this article includes the following source data and figure supplement(s) for figure 1:

**Source data 1.** Source Data for *Figure 1A*.
**Source data 2.** Source Data for *Figure 1B*.
**Figure supplement 1.** Examples of flow cytometry and S-phase labeling.
**Figure supplement 2.** Polyploidy in Optic lobes is not light or photoreceptor dependent.

remains variable from animal to animal, but on average, does not increase. We observe similar patterns of polyploidy accumulation in males and females (*Figure 1A*). To ensure that the polyploidy we observe is not an artefact of one particular strain, we performed similar measurements across the lifespan in other commonly used lab 'wild-type' strains *Canton-S* (*Figure 1B*) and *Oregon-R* (*Figure 1C*). Interestingly, *Oregon-R* flies show lower levels of polyploidy in the first two weeks than *w1118* and *Canton-S* suggesting that different genetic backgrounds may influence polyploidy in the brain. We also performed DNA content measurements of brains from the distantly related *D.americana* which diverged ~50 million years ago and a more closely related species, *D.mauritiana*, which diverged ~2 million years ago (*Figure 1C*). While both species show accumulation of polyploidy, it is interesting to note that they show differences in levels of polyploidy.

We next measured changes in ploidy in the *D. melanogaster* adult brain over time. We pooled data from multiple animals and binned polyploid cells from $w^{1118}$ brains into two categories: cells with 4C (tetraploid) DNA content measured by flow cytometry and cells with >4C DNA content - this includes 8C, 16C and even some 32 C cells (*Figure 1D*). The majority of the polyploid cells appear to be tetraploid, and the fraction of cells exhibiting >4C DNA content increases during the first week of adulthood, but remains relatively consistent with age.

We next asked whether polyploid cells are located in a specific region of the brain. We dissected the *Drosophila* central nervous system (CNS) into the central brain, optic lobes, and ventral nerve cord (VNC) and measured levels of polyploidy in each region from day of eclosion to 2 weeks into adulthood (*Figure 1E*). We found that while there is a low level of polyploidy in the central brain and VNC that increases with age, most of the polyploidy comes from the optic lobes. Strikingly, by 3 weeks, over 20% of the cells in the optic lobes can exhibit polyploidy.

Since the optic lobes contribute to most of the polyploidy observed, we wondered if this phenomenon may be dependent on light. *Canton-S* animals reared in complete darkness did not show difference in polyploidy compared to age-matched controls raised in regular 12 hr light/12 hr dark cycles (*Figure 1—figure supplement 2A*). Next, we hypothesized that polyploidy accumulation may depend on proper photoreceptor function. However, $glass^{60j}$ flies devoid of photoreceptors and pigment cells in compound eyes (*Helfrich-Förster et al., 2001*) still show polyploidy (*Figure 1—figure supplement 2B*).

## Multiple cell types exhibit adult-onset polyploidy in the brain

To identify which cell types in the brain are becoming polyploid, we used the binary GAL4/UAS system to drive the expression of a nuclear-localised green or red fluorescent protein (nGRP or nRFP) with cell type-specific drivers. We then measured DNA content using Dye-Cycle Violet in the GFP or RFP-positive populations.

We first examined the SPGs, as previous work from the Orr-Weaver lab identified these to be highly polyploid (*Unhavaithaya and Orr-Weaver, 2012*; *Von Stetina et al., 2018*). When we used the SPG driver *moody-GAL4*, we found that the SPGs are highly polyploid (*Unhavaithaya and Orr-Weaver, 2012*), but contributed to less than 5% the polyploid cells observed in mature adult brains (*Figure 2—figure supplement 1A,C*). Another class of cells previously shown to be polyploid in some contexts are tracheal cells that carry oxygen to tissues (*Zhou et al., 2016*). Using the tracheal driver *breathless-GAL4*, we found that in 10 day old adult brains, tracheal cells comprise less than 3% of all polyploid cells (*Figure 2—figure supplement 1B,C*). Thus, 90% of the polyploid cells we observe in the brain arise from cell types not previously known to become polyploid.

The adult fly brain is thought to be composed almost entirely of neurons (90% of total population) and glia (10% of total population). First, we asked if neurons become polyploid by using a pan-neuronal driver, *nSyb-GAL4* to drive *UAS-nGFP* (*Figure 2A*). We found that indeed, by two weeks ~ 5–6% of cells expressing nSyb-GAL4 show >2C DNA content. Similarly, we used the pan-glial driver *Repo-GAL4* and found that by 2 weeks ~ 6–7% of glia also become polyploid in the adult brain (*Figure 2B*). Neurons outnumber glial cells in the fly brain, and we find that the relative proportions of polyploid cells reflect the total ratio of neurons vs. glia in the adult brain (*Figure 2C*). We next asked whether specific types of neurons or glia show higher levels of polyploidy. We measured the proportion of polyploid vs diploid cells in various classes of neurons (*Figure 2D*) and glia (*Figure 2E*) in 7 day old optic lobes. Interestingly, we found that most differentiated cell types we assayed in the optic lobes show some level of polyploidy by one week of age. We conclude that polyploidy arises in multiple neuronal and glial types that are initially diploid upon eclosion and become polyploid after terminal differentiation and specifically during adulthood.

## Most of the polyploidy is not a result of cell fusion

We reasoned that cells in the brain could become polyploid either by re-entering the cell cycle or by undergoing cell fusion (*Alvarez-Dolado and Martínez-Losa, 2011*; *Giordano-Santini et al., 2016*; *Grendler et al., 2019*; *Losick et al., 2013*; *Schoenfelder et al., 2014*; *Shu et al., 2018*; *Starnes et al., 2016*). To examine whether cell fusion occurs, we used a genetic labeling tool called CoinFLP (*Bosch et al., 2015*). The CoinFLP genetic cassette contains two overlapping but exclusive Flippase Recombination Target (FRT) sites flanking a stop cassette that can be 'flipped -out' using

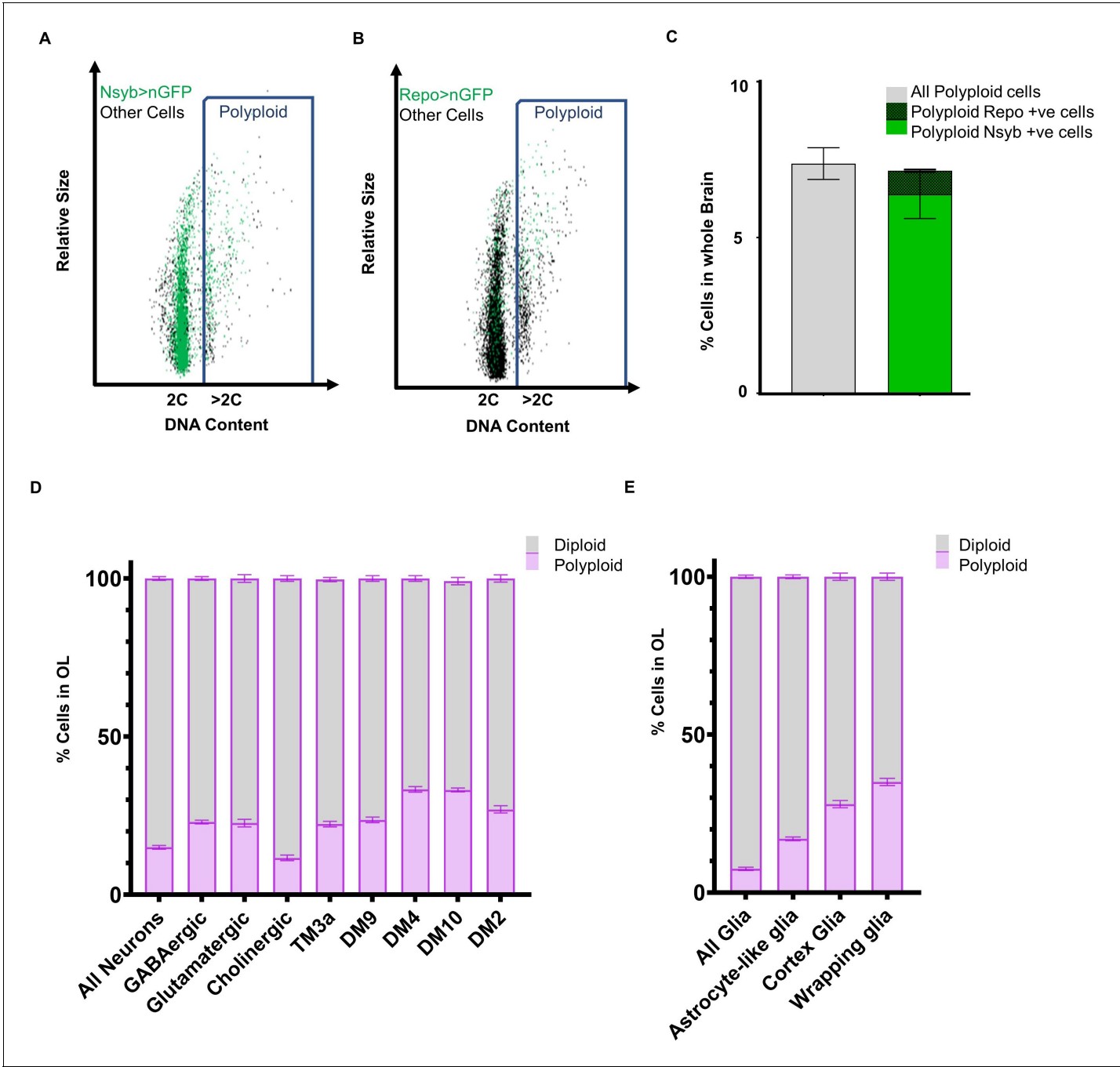

**Figure 2.** Identification of various neuronal and glial cell types that become polyploid in the adult brain. Representative flow cytometry dot plots showing polyploid neuronal (A) and glial (B) cells in 2 week old male brain (A) Neuronal nuclei are labeled using *nsyb-GAL4, UAS-nGFP*, neuronal cells are shown in the dot plot as green dots and 'other' cells unlabelled by *nsyb-GAL4* are shown in black. Blue rectangle highlights cells with polyploid or >2C DNA content (B) Glial nuclei are labelled using *Repo-GAL4, UAS-nGFP*, glial cells are shown in the dot plot as green dots and 'other' cells unlabeled by Repo-GAL4 are shown in black. Blue rectangle highlights cells with polyploid or >2C DNA content. (C) Plot showing proportion of polyploid neurons (bold green) and polyploid glia (checked green) at 2 weeks compared to total polyploidy in the brain in $w^{1118}$ control (grey) (error bars show mean ± SEM, n = 3). (D) Proportion of polyploidy observed at 7 days in the optic lobes in various classes of neurons (D) and glia (E). Stacked bar plot showing mean ± SEM; percentage of polyploidy (purple) and diploidy (grey) per sample, each sample contains pooled OLs from three or more brains; n = 3. Proportions of cells also represented as tables in *Supplementary files 3* and *4*.

The online version of this article includes the following source data and figure supplement(s) for figure 2:

**Source data 1.** Source Data for *Figure 2C*.
**Source data 2.** Source Data for *Figure 2D*.

*Figure 2 continued on next page*

*Figure 2 continued*

**Source data 3.** Source Data for *Figure 2E*.
**Figure supplement 1.** Trachea and Sub-perineurial glia comprise less than 5% of all polyploid cells.

FRT mediated recombination to give rise to cells expressing either a LexGAD driver or a GAL4 driver, which can be used to drive expression of *lexA$_{op}$-GFP* (green) and *UAS-RFP* (red). In animals heterozygous for CoinFLP, a diploid cell has only one copy of the transgenic cassette which can only be 'flipped' to give rise to a cell permanently labeled with either red or green fluorescent proteins, hence the name CoinFLP. If labeling is induced in the brain early during development before eclosion, cells become stochastically and permanently labeled with either red or green fluorescent proteins. If cells fuse in the ageing brain, up to ⅓ of cells undergoing fusion could fuse a red-labeled cell with a green cell and appear yellow. We used a FLP recombinase (flippase) under the control of the *eyeless* promoter (ey-FLP) to label most of the cells red or green in the optic lobes early in development (*Figure 3A–B'*) and did not observe any double labeled (yellow) cells in young adult brains or older brains. We also expressed flippase enzyme more broadly under the control of a heat-shock promoter (hs-FLP) and labeled cells using a nuclear GFP or RFP at larval L2-L3 stages (*Figure 3C*) and measured the number of double-labeled cells in the optic lobes at on the day of eclosion or after aging at 14 days post-eclosion. We never observed more than 10–12 cells per optic lobes exhibiting double-labeling under these 'early-FLP' conditions. These double-labeled cells under the larval hs-FLP conditions likely include the SPG cells which become polyploid early in larval development and do not express ey-FLP. We conclude that very little cell fusion occurs in the adult OL, even with age.

## 'Late-FLP' can label polyploid cells that arise from cell cycle reentry

By using a modified labeling paradigm, we can also use CoinFLP to label polyploid cells in situ (*Figure 4A*). Previous work with CoinFLP has shown that inducing 'flipping' in cells that are already polyploid results in a fraction of double labeled yellow cells (*Bosch et al., 2015*). We therefore reasoned that heterozygous CoinFLP brain cells that become polyploid by replicating their genome during adulthood will contain two or more copies of the CoinFLP transgene cassette. If we label cells by activating *hs-FLP* late in adulthood after polyploidy appears, some polyploid cells may 'flip' one copy green and one copy red, appearing yellow. When we induce an adult FLP at one day, before polyploidy occurs, we do not observe any double-labeled cells in the optic lobe (*Figure 4B*) but when we induce an adult FLP at 30 days post-eclosion, we observe several double labeled cells (*Figure 4B'*) indicating that these cells have undergone genome replication and contain at least two heterozygous copies of the CoinFLP transgenic cassette. To quantify this, we used an adult 'late-FLP' to drive nuclear GFP and RFP, and we observe around 300 double-labeled cells per optic lobes in 14 day old brains (*Figure 4C*). The presence of double-labelled nuclei in aged optic lobes suggest cells become polyploid by cell cycle re-entry and endoreduplicating DNA.

We further confirmed that the double-labeled cells visualized by microscopy are polyploid by performing DAPI intensity quantification in high-magnification images (*Figure 4—figure supplement 1*) and flow cytometry. (*Figure 4—figure supplement 2*). All CoinFLP double-labeled cells were confirmed to be polyploid by DAPI integrated intensity measurements and 97% of all double-labeled cells show >2C DNA content by FACS (*Figure 4—figure supplement 2*). CoinFLP double-labeling confirmed several types of glia identifiable by location and shape to become polyploid, such as a subset of cortex glia of the outer (*Figure 4D*) and inner (*Figure 4E*) optic chiasm and astrocyte-like glia (*Figure 4F*) in the medulla of the OL.

To test whether polyploidy in the adult optic lobes is driven by cell cycle re-entry, we used cell-type specific RNA-interference (RNAi) to modulate the DNA replication licensing factors cdc6 and Geminin in postmitotic neurons. Cdc6 is an essential factor for DNA replication licensing that promotes the recruitment of the MCM complex to load the DNA replication complex (*Kang et al., 2014*), while Geminin is a replication licensing inhibitor that sequesters DNA replication licensing factors to inhibit DNA re-replication (*Lutzmann et al., 2006*). Using *nSyb-GAL4*, we expressed *UAS-cdc6$^{RNAi}$* in differentiated neurons which significantly reduced levels of polyploidy by 14d (*Figure 4G*) from ~25% in control optic lobes to ~14% on optic lobes expressing the RNAi. We next

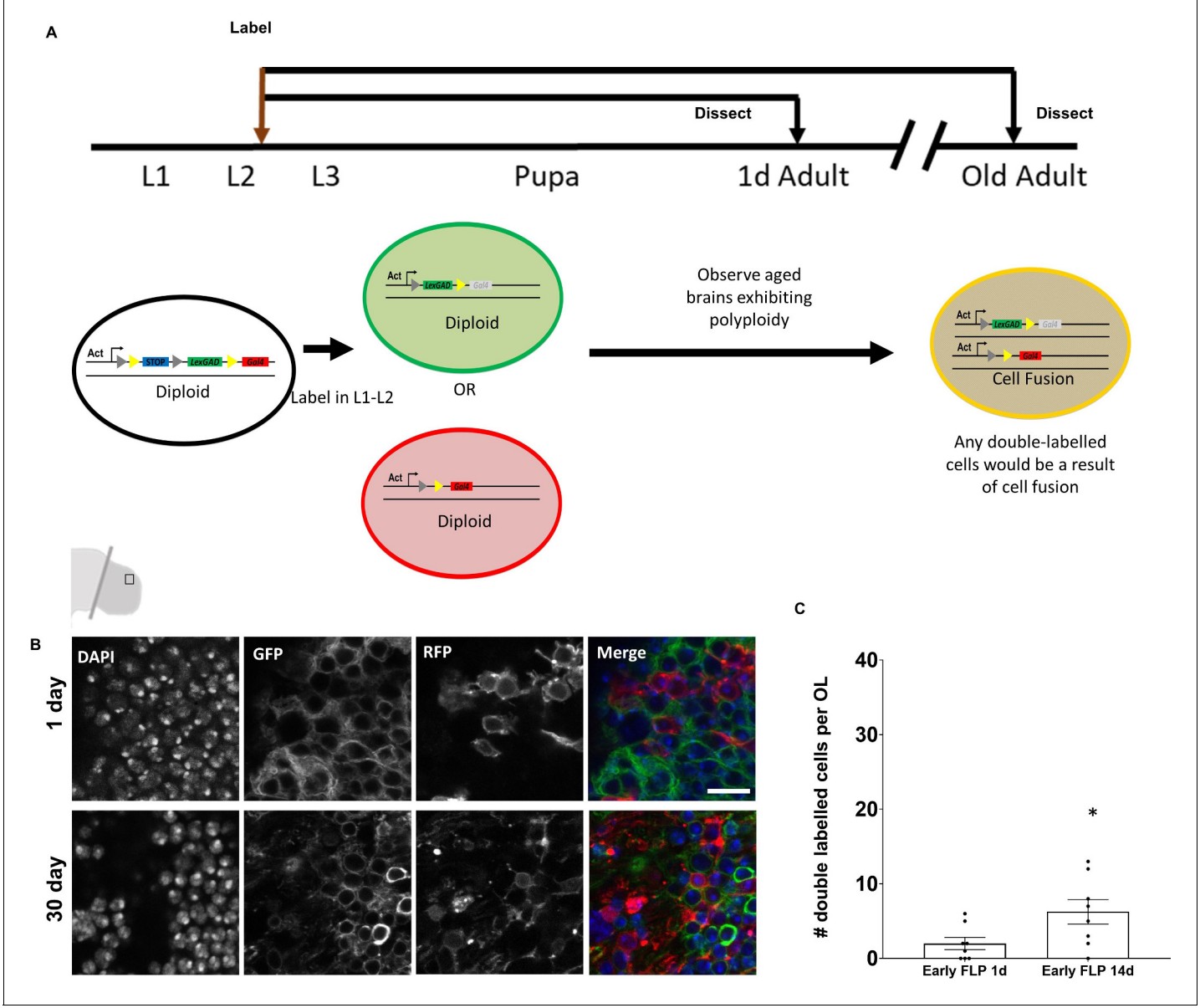

**Figure 3.** Very few polyploid cells arise from cell fusion in the adult brain. (**A**) Schematic of 'early labeling' using CoinFLP to identify potential cell fusion events. Early CoinFLP labeling will label diploid cells either with GFP or RFP. Any double-labeled cells in an older, polyploid brain will be a result of cell fusion. Representative images of 0 day (**B**) and 30 day polyploid optic lobes showing no double-labeled cells under 'Early-FLP' conditions when labeled using *ey-FLP* and membrane GFP and RFP. (**C**) Quantification of double labeled cells using nuclear GFP and RFP observed per brain lobe in early 'FLP' condition at 14 days. p value=0.0428 significance calculated using unpaired t-test with Welch's correction. Early 'FLP' in (**C**) was induced at L2-early L3 stages using *hs-FLP*. Scale bars = 8.3 μm.

knocked down geminin and found that we increase levels of polyploidy in the optic lobes (*Figure 4G*). This suggests that a fraction of post-mitotic neurons reactivate DNA replication to become polyploid in the adult fly brain.

## DNA damage accumulates in adult optic lobes

To investigate whether transcriptional changes that occur with age may be associated with cell cycle reactivation and polyploidy in the brain, we performed RNA sequencing on three parts of the CNS: optic lobes, central brain and VNC from male and female *Canton-S* animals at different time points:

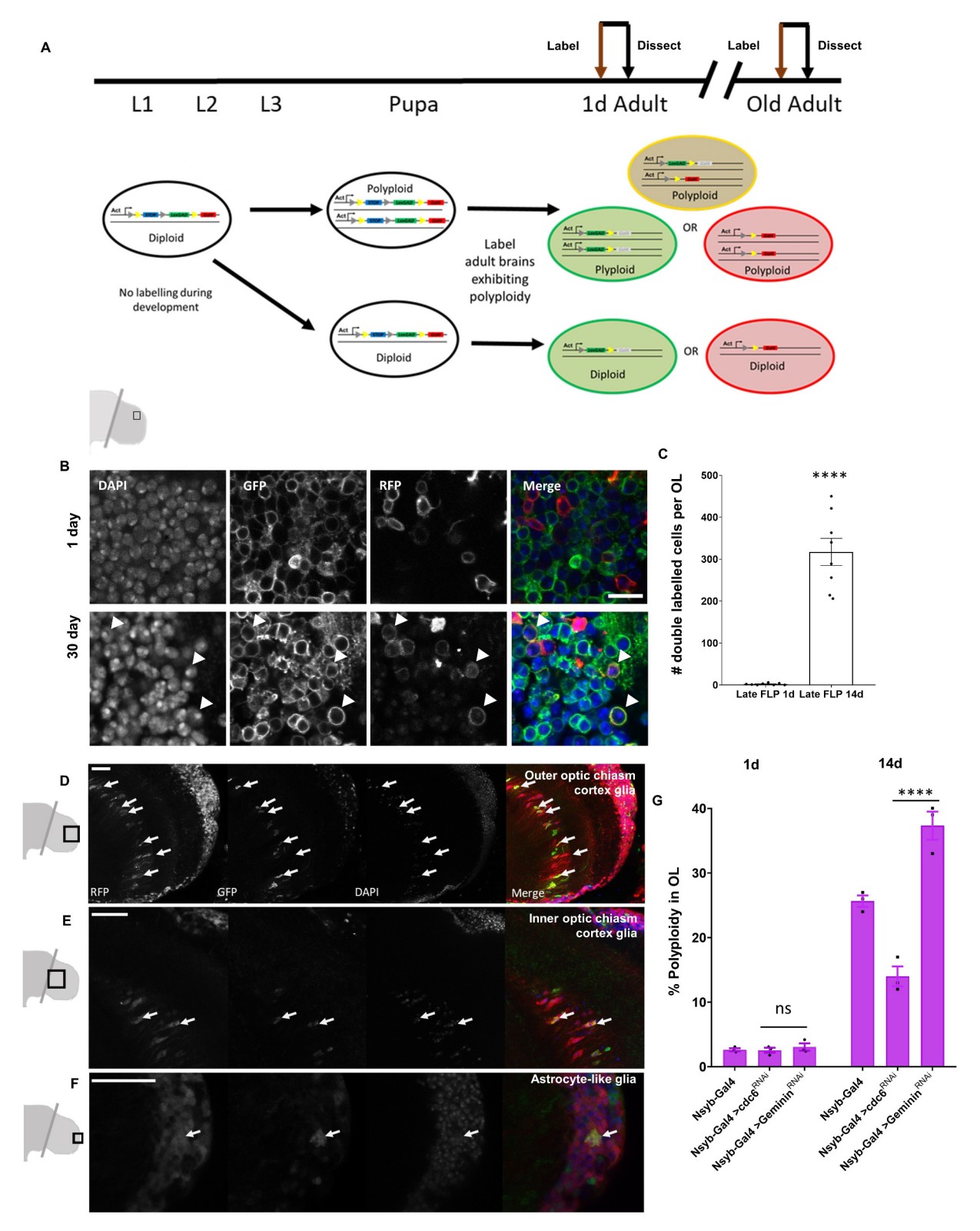

**Figure 4.** Cells in the Optic Lobes undergo cell cycle re-entry to become polyploid. (**A**) Schematic showing labeling protocol for inducing a 'late-FLP' in brains where polyploidy is expected to identify polyploid cells in situ. A proportion of cells with multiple copies of the genome will be double-labeled. Representative images of 1 day optic lobe heat shocked soon after eclosion and a 30 day old optic lobe heat shocked at 29 days to induce labeling (**B**). Older optic lobes have double-labeled cells marked with membrane GFP and RFP. Scale bar = 8.3 μm. (**C**) Quantification of double labeled cells using

*Figure 4 continued on next page*

*Figure 4 continued*

nuclear GFP and RFP observed per brain lobe in 'late-FLP' condition. Labeling was induced 24 hr prior to dissection for both 1d and 14d using *hs-FLP*. P value < 0.0001 significance calculated using unpaired t-test with Welch's correction. (**D–F**) Representative images showing cortex glia of the outer (**D**) and inner (**E**) optic chiasm as well as astrocyte-like (**F**) glial nuclei that can be identified as polyploid based on position and morphology using CoinFLP 'late-FLP' labeling method. Polyploid, double-labeled glia of each type are indicated with white arrows (**G**) Inhibition of DNA replication licensing factor *cdc6* by RNAi in neurons using the driver *nsyb-GAL4* results in lower levels of polyploidy (measured by flow cytometry) in male optic lobes compared to control (GAL4 driver alone). Knockdown of replication inhibitor *geminin* increases levels of polyploidy in 14 day old male optic lobes. Error bars show mean ± SEM, n = 3. (Two way anova with greenhouse geisser correction for unequal SDs followed by Holm-Sidak's multiple comparisons test p values: 0.1234 = ns;<0.0332 *;<0.0021 **;<0.0002 ***; ****<0.0001) Scale bars for D-F = 20 μm.

The online version of this article includes the following source data and figure supplement(s) for figure 4:

**Source data 1.** Source Data for *Figure 4C*.
**Figure supplement 1.** CoinFLP double positive cells are polyploid.
**Figure supplement 2.** CoinFLP 'double-labeled' cells have polyploid DNA content.
**Figure supplement 2—source data 1.** Source Data for *Figure 4—figure supplement 2* panel A.

1 day, 2 days, 7 days and 21 days post-eclosion. To infer biological processes that are affected with age, gene ontology analysis was performed using GOrilla and redundant terms were filtered using reviGO. The most significant changes observed in the optic lobes at 21d compared to 2d are shown in *Figure 5A,B*. Among the most significantly upregulated groups of genes are those associated with the cell cycle, DNA damage response and DNA damage repair. The enrichment of up-regulated genes associated with the DNA damage response was also observed in the optic lobes at 7 days (*Figure 5—figure supplement 1*), but the enrichment and fold-induction of specific genes is stronger at day 21 (*Figure 5A*). A gene expression signature associated with DNA damage is specific to the optic lobes. However, the most significantly downregulated GO terms in the optic lobes at 21d are shared with the central brain and VNC and include metabolism, transmembrane transport and cellular respiration-associated processes (*Figure 5—figure supplement 1*).

To examine whether DNA damage is higher in the OL, we performed immunostaining against the phosphorylated histone 2A variant (pH2AV) in 1d optic lobes and central brain and 21d optic lobes and central brain (*Figure 5C–E*) from *Canton-S* male brains. Young brains show very low levels of pH2AV in both the optic lobes and central brain (*Figure 5C,C',E*) but older brains show higher levels of pH2AV in the optic lobes compared to the central brain (*Figure 5D,D',E*).

To further understand the DNA damage and cell cycle signatures observed with age, we looked at the change in expression of specific genes involved in the DNA damage response and the cell cycle (*Figure 5F*). Recent work has identified a specific transcriptional response to induced DNA damage in the head that involved a non-canonical role for tumor suppressor protein p53 (*Kurtz et al., 2019*). This signature was termed head Radiation Induced p53-Dependent or hRIPD. In addition to genes such as *FANCI*, *loki*, *rad50* and *xrp1* which are involved in a canonical, ionising radiation-induced DNA damage response, we also find robust upregulation of hRIPD genes *Ku80*, *Irbp*, *Cht8*, *CG3344* and *CG4734* in our RNAseq data set at 7d and 21d in optic lobes. However, these genes are not as strongly upregulated in the aged central brain or VNC and *p53* itself shows only a small increase in the optic lobes at 21d (*Figure 5F*). Consistent with cell cycle re-entry in a fraction of cells in the OL, upregulation of cell cycle genes such as *myc*, *cyclin D*, *orc1* is observed specifically in the optic lobes and increases with age.

## Polyploidy accumulation in neurons is p53-independent

Work in other polyploid tissues in *Drosophila* has shown that polyploid cells in the salivary gland can tolerate high levels of DNA double-strand breaks and resist apoptosis caused by DNA damage (*Hassel et al., 2014*; *Mehrotra et al., 2008*; *Qi and Calvi, 2016*; *Zhang et al., 2014*). This is possible because polyploid cells in tissues such as the salivary gland have intrinsically low levels of p53 protein and also suppress the expression of pro-apoptotic genes (*Zhang et al., 2014*). It has also been shown in various tissues and organisms that DNA damage can induce polyploidization (*Bretscher and Fox, 2016*; *Donovan and Corbo, 2012*; *Grendler et al., 2019*). Since we observe a modest upregulation of p53 as well as a p53-dependent gene expression signature in older optic lobes, we asked if the induction of polyploidy in neurons is p53 dependent. To address this, we over-expressed wildtype (p53^WT^) or a dominant-negative allele of p53 (p53^DN^) that is unable to bind to

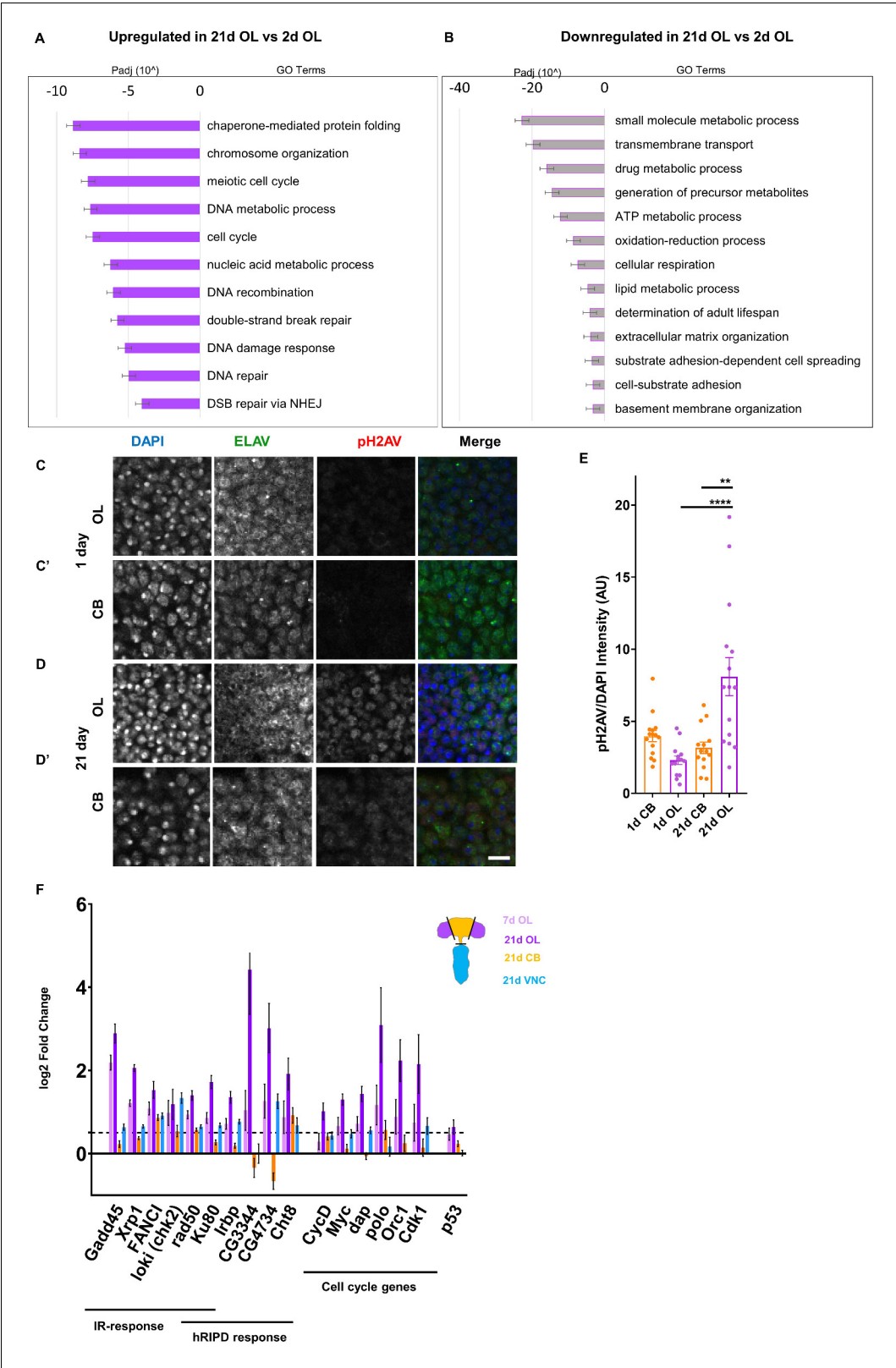

**Figure 5.** DNA damage accumulates with age in the Optic Lobes Changes in gene expression in 21d OL compared to 2d OL shown by GO term analysis. Padj = adjusted P value. Upregulated GO terms shown in solid purple (**A**), downregulated GO terms shown in grey bars outlined with purple (**B'**). Representative images showing pH2AV foci in 1 (**C,C'**) and 21 day (**D,D'**) Central Brain (CB) and Optic Lobes (OL) Neurons are labeled in green (ELAV), phosphorylated histone 2A variant (pH2AV) in red nuclei are labeled in blue (DAPI). (**E**) Accumulation of DNA damage is quantified by

*Figure 5 continued on next page*

*Figure 5 continued*

measuring pH2AV intensity/DAPI intensity per frame in five brains per sample. Significance determined by performing unpaired t-test with Welch's correction for unequal SD. Scale bars = 20 μm. (F) Genes involved in canonical Ionising Radiation (IR) response, head radiation induced p53 dependent (hRIPD) and cell cycle genes showing changes in expression compared to 2 day. Dotted line indicates threshold for significance. Genes showing changes in 7d OL are shown in light purple, 21d OL are shown in dark purple, 21D CB in yellow and 21d VNC in blue.

The online version of this article includes the following source data and figure supplement(s) for figure 5:

**Figure supplement 1.** Supplemental RNAseq GO analysis.
**Figure supplement 2.** Validation of RNAseq data Validation of our RNA sequencing dataset was performed by comparing our dataset to other published datasets.
**Figure supplement 2—source data 1.** The list of comparisons and references for validation of RNAseq dataset.

DNA and evoke a transcriptional response in neurons using the *nSyb-GAL4* driver (*Figure 6A*). We did not see a significant difference in levels of polyploidy in 7 day old brains with overexpression of either WT or mutant p53, suggesting that accumulation of polyploidy in neurons is p53-independent. We also performed cell death measurement in the brain using flow cytometry. We calculated cell death by measuring proportions of cells incorporating either Propidium Iodide (PI) or Sytox-Green (*Figure 6—figure supplement 1*). We did not see a significant difference in the proportion of dead cells in overexpression of p53$^{WT}$ or p53$^{DN}$ conditions compared to control (*Figure 6—figure supplement 1B*) consistent with recent work suggesting a non-canonical, non-apoptotic role for p53 in the adult *Drosophila* brain (*Kurtz et al., 2019*).

## Exogenous DNA damage leads to increased polyploidy

We next asked if exogenous stress can impact levels of polyploidy in the brain. Increased oxidative stress is commonly associated with ageing (*Haddadi et al., 2014*; *Hussain et al., 2018*; *Pinto and Moraes, 2015*). We first treated flies with a low dose of paraquat (PQ) to mimic oxidative stress (*Bonilla et al., 2006*; *Dudas and Arking, 1995*; *Hosamani and Muralidhara, 2013*; *Zou et al., 2000*). $w^{1118}$ flies treated with low dose of 2 mM PQ from eclosion show increased DNA damage as well as increased polyploidy at 7d −14d (*Figure 6B,D*) but not increased cell death (*Figure 6—figure supplement 1*).

We next tested whether inducing DNA damage directly affects polyploidy. We treated flies with 900mJ of UV radiation by placing flies in a UV Stratalinker at 2 days post-eclosion (*Grendler et al., 2019*; *Kang and Bashirullah, 2014*) and observed significantly increased levels of polyploidy at day seven in UV-treated flies compared to mock-treated controls (*Figure 6C*). We measured cell death using propidium-iodide (PI) incorporation (*Grushko and Buttitta, 2015*) and observed an acute increase in cell death 16 hr post-exposure to UV (*Figure 6E*), but no difference in cell death 5 days post-exposure. This suggests that cell death precedes accumulation of polyploidy upon induction of exogenous DNA damage.

## Polyploid cells are protected from cell death

Polyploid cells in other tissues are known to sustain high levels of DNA damage as well as resist cell death (*Zhang et al., 2014*). We and others do not observe reproducible caspase-dependent cell death in the adult brain beyond the first 5 days after eclosion (*Foo et al., 2017*). We measured cell death and necrosis in individual $w^{1118}$ adult brains over a time-course using PI incorporation from day one post-eclosion until day 56 (*Grushko and Buttitta, 2015*). We found that newly eclosed flies exhibit a low level of dead or dying cells but from day 7 to day 56, the brain shows a relatively steady level (~5%) of dead or dying cells (*Figure 6F*) although there is variability from animal to animal. Since dead cells are cleared in the brain (*Kurant, 2011*), we expect this steady rate of cell death to result in a predictable rate of cell loss in the brain with age, which closely agrees with our total cell counts performed using flow cytometry (*Figure 6—figure supplement 1*).

We next examined whether the polyploid cells in aged brains are protected from cell death. Since the numbers of dead or dying cells measured in individual brains was very small, we pooled brains from 2 week old *Canton-S* males to obtain a measurement of ploidy in the PI positive cells by co-staining with the DNA content dye DyeCycle Violet. We found that while ~7% of the diploid cells

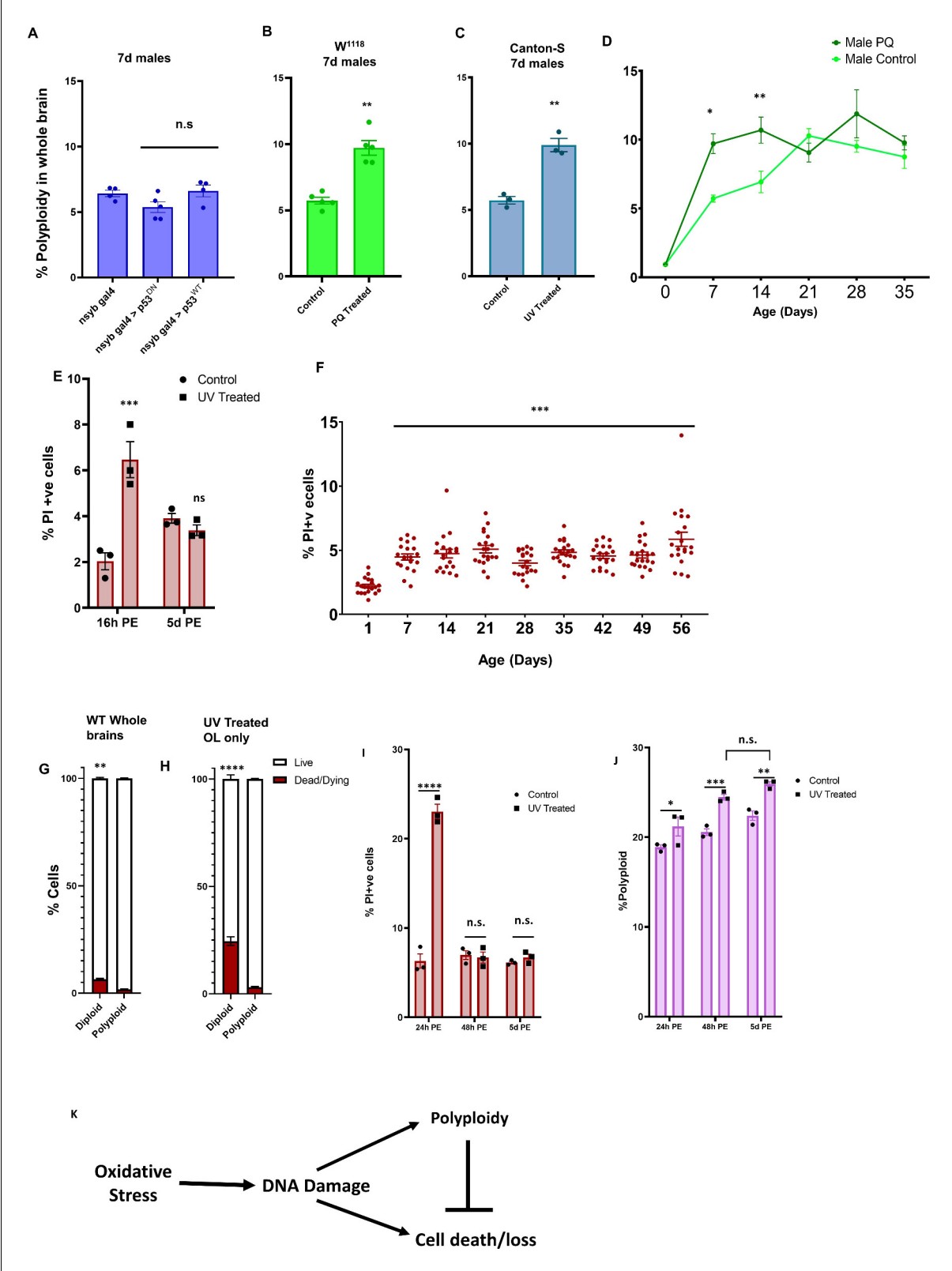

**Figure 6.** Oxidative stress and DNA damage results in increased polyploidy during early adulthood and polyploid cells are protected from cell death.
(**A**) Polyploidy in neurons is not p53 dependent. Percentage of polyploidy under each condition was measured in individual 7d male brains (n = 5). (**B**) $w^{1118}$ males treated with 2 mM paraquat (PQ) from day of eclosion exhibit higher levels of polyploidy at 7 days compared to control $w^{1118}$ males (n = 5).
(**C**) UV treated (960mJ exposure 5 days prior to dissection and flow cytometry) *Canton-S* flies show greater levels of polyploidy at 7 days compared to
*Figure 6 continued on next page*

*Figure 6 continued*

control (n = 3). Error bars are mean ± SEM, significance was calculated by performing unpaired t-test with Welch's correction for unequal SD. (**D**) Accumulation of polyploidy over a time course in $w^{1118}$ males on 2 mM PQ (dark green) compared to control $w^{1118}$ males (light green). Shapes show mean, bars show SEM. Significance was calculated using two way ANOVA with Greenhouse-geisser correction for unequal SDs, multiple comparisons with Holm-Sidak's test; 0.1234 = ns;<0.0332 *;<0.0021 **;<0.0002 ***. (**E**) Cell death measured by Propidium Iodide incorporation in animals treated with 960mJ UV at 16 hr post-exposure and 5 days post-exposure. Cell death precedes accumulation of polyploidy upon induced DNA damage. Significance was calculated using two way ANOVA with Greenhouse-geisser correction for unequal SDs, multiple comparisons with Holm-Sidak's test; 0.1234 = ns;<0.0332 *;<0.0021 **;<0.0002 ***; ****<0.0001. (**F**) PI incorporation shows percentage of dead/dying cells in individual brains, male and female, $w^{1118}$ at different ages post-eclosion. Significance was calculated using two way ANOVA with Greenhouse-geisser correction for unequal SDs, multiple comparisons with Holm-Sidak's test; 0.1234 = ns;<0.0332 *;<0.0021 **;<0.0002 ***; ****<0.0001 (**G**). Proportion of PI+ cells that are diploid (2C) and polyploid (>2C) in pooled 14 day old *Canton-S* male brains. (**H–J**) Animals were exposed to 480 mJ UV at 21 days and dissected 24 hr, 48 hr or 5d post exposure and cell death (**H,I**) and polyploidy (**J**) was measured by flow cytometry. (**H**) Proportion of PI+ cells that are diploid (2C) and polyploid (>2C) in pooled 21d $w^{1118}$ male OL 24 hr post-exposure to 480mJ UV. (**K**) Proposed Model. For (**G–I**) n = 3, Error bars are mean ± SEM, significance was calculated by performing Two way ANOVA, multiple comparisons with Holm-Sidak's test; 0.1234 = ns;<0.0332 *;<0.0021 **;<0.0002 ***; ****<0.0001. The online version of this article includes the following source data and figure supplement(s) for figure 6:

**Source data 1.** Source Data for *Figure 6D*.
**Source data 2.** Source Data for *Figure 6F*.
**Source data 3.** Source Data for *Figure 6H*.
**Source data 4.** Source Data for *Figure 6I*.
**Source data 5.** Source Data for *Figure 6J*.
**Figure supplement 1.** Supplemental Cell death and DNA damage data.

incorporate PI, less than ~2.5% of polyploid cells incorporate PI (*Figure 6G*), suggesting that polyploid cells are more resistant to cell death.

To examine whether polyploid cells are resistant to cell death upon external DNA damage, we aged animals to 21 days, a time point where the OL exhibits high levels of polyploidy. We then exposed these flies to 480mJ UV to induce DNA damage and measured the levels of cell death and polyploidy from 24 hr – 5 days post exposure to UV. We observe high levels of PI incorporation (*Figure 6H,I*) at 24 hr post exposure, but normal levels by 48 hr, indicating an acute response of DNA damage induced cell death in the brain. Many diploid cells die in response to this dose of UV (~24.5%). In contrast, the polyploid cells show very low levels of PI incorporation (~3.1%, *Figure 6H*), suggesting that the polyploid cells in older adult brains are also resistant to DNA damage induced cell death. We next examined whether the exposure to DNA damage also altered polyploidy, as we had observed in younger animals (*Figure 6C*). At 48 hr we observed, on average, a 4% increase in polyploidy for UV exposed animals. This early increase can be almost entirely attributed to the loss of the diploid cells that are PI positive at 24 hr post exposure (a loss of 24.5% of diploid cells increases the proportion of polyploid cells from 18.9% to 23%). When comparing 48 hr to 5 days post exposure, we see no significant increase in polyploidy, suggesting that after 3 weeks of age, animals lose the ability to further increase polyploidy in response to damage. This is in contrast to our experiment in young animals, using a low dose of paraquat to cause oxidative damage (*Figure 6B,D*) where we see an earlier accumulation of polyploid cells without any obvious increase in cell death (*Figure 6—figure supplement 1*).

The work described in this study supports a model (*Figure 6K*) where cells in the early adult fly brain undergo endoreplication and polyploidization in response to DNA damage and oxidative stress accumulated with age. Our data also suggest that polyploid cells are more resistant to cell death and may serve a beneficial or neuroprotective role in the ageing brain.

## Discussion

### Adult-onset polyploidy in neurons and glia

In this study we describe a surprising discovery, that diploid cells in the adult *Drosophila* brain can re-enter the cell cycle and become polyploid. We have identified several classes of neurons as well as glia that exhibit adult-onset polyploidy. We have also characterized which regions of the brain

show increased polyploidy, and find that polyploidy is closely correlated with the expression of a DNA damage response signature. Other work has also shown that a small population of about 40 stem cells in the optic lobes of *Drosophila* respond to acute injury by generating adult-born neurons (*Fernández-Hernández et al., 2013*). We considered the possibility that a fraction of cells with 4C DNA content may be in G2 and poised to undergo mitosis. We stained for G2 and mitotic cell cycle markers (phospho-histone H3 and Cyclin A) in over 100 adult brains at different ages and never observed a convincing G2 or mitotic event. However, we may have missed rare, transient cell cycle events that are captured by permanent lineage tracing approaches (*Crocker et al., 2020*; *Fernández-Hernández et al., 2013*). Both our cell number counts and cell death measurements indicate a steady decline in cell number in the adult brain with age (*Figure 6—figure supplement 1*), and suggest that under normal ageing conditions mitoses are likely rare. Moreover, we observe hundreds to thousands of tetraploid or polyploid cells by FACS or CoinFLP, suggesting that only a very small proportion of tetraploid cells would be expected to be in G2. We suggest that multiple mechanisms are employed in this brain region to ensure proper function and tissue integrity with age.

Polyploidy in neurons has previously been reported in the mouse cerebral cortex (*López-Sánchez et al., 2017a*; *López-Sánchez and Frade, 2013*) and chick retinal ganglion cells (*Morillo et al., 2010*). Whether purkinje cells in the mammalian cerebellum are polyploid has been a matter of considerable debate over the past several decades. (*Brodskii et al., 1971*; *Del Monte, 2006*; *Kemp et al., 2012*; *Lapham, 1968*; *Lapham et al., 1971*; *Mares et al., 1973*; *Swartz and Bhatnagar, 1981*). Perhaps the most exaggerated examples of polyploidy are from the giant neurons in the terrestrial slug *Limax* (*Yamagishi et al., 2011*) and the sea slug *Aplysia* (*Coggeshall et al., 1970*) where giant neurons contain >100,000 copies of the diploid genome. However, in all these cases, polyploid neurons appear during development. Our study describes a novel phenomenon of adult onset and accumulation of polyploidy in the *Drosophila* brain under normal physiological ageing conditions.

## What is the function of polyploidization in the brain?

We have shown that many cell types become polyploid in the adult brain (*Figure 2*). These cell types have distinct physiology and functions. How polyploidization affects the function of these various cell types is an exciting avenue for future research. Polyploidy can confer cell-type and context specific benefits in various tissues. In *Drosophila*, polyploid intestinal enterocytes (*Miguel-Aliaga et al., 2018*), SPGs (*Unhavaithaya and Orr-Weaver, 2012*; *Von Stetina et al., 2018*) and cells in the wounded epithelium (*Losick, 2016*; *Losick et al., 2013*) undergo endoreduplication and do not undergo cytokinesis to maintain the integrity of the blood brain barrier and the cell-cell junctions in the epithelium respectively. One possibility is that polyploidy in neurons or glia may allow cells to compensate for cell loss while maintaining established cell-cell contacts (*Unhavaithaya and Orr-Weaver, 2012*). The compound eye and optic lobes of *Drosophila* contain ~750–800 ommatidial 'units' that form a highly organized and crystalline structure (*Bates et al., 2019*; *Nériec and Desplan, 2016*; *Pecot et al., 2014*). Numerically and topographically matched cells in the medulla cortex of the optic lobes receive inputs from the lamina which in turn receives inputs from the retina (*Bates et al., 2019*; *Pecot et al., 2014*). We observe polyploidization in multiple neuronal types found in the medulla, yet several cell types in the brain show a decline in number with age (*Bates et al., 2019*). In neurons, polyploidy could play a role in helping cells increase their soma size and dendritic arbors (*Morillo et al., 2010*; *Szaro and Tompkins, 1987*). It is possible that polyploidy allows neurons to form more presynaptic and postsynaptic connections to compensate for lost cells while maintaining the integrity of existing connections in the visual system.

Nurse cells in the egg chamber (*Lilly and Spradling, 1996*; *Wattiaux and Tsien, 1971*), cells in the accessory gland (*Box et al., 2019*; *Sitnik et al., 2016*), salivary gland (*Edgar and Orr-Weaver, 2001*) and fat body (*Guarner et al., 2017*), on the other hand become polyploid to fulfill increased biosynthetic demands. In addition to an upregulation of DNA damage and cell cycle in the optic lobes, our RNAseq data suggest compromised metabolism with age in all parts of the brain. One of the main functions of glial cells is to provide metabolic support to neurons in the brain (*Kremer et al., 2017*; *Schirmeier et al., 2016*; *Volkenhoff et al., 2015*). Polyploidization in astrocyte and cortex glial cells might also serve to increase their metabolic output and compensate for the reduced metabolic output in the ageing brain.

## DNA damage accumulates in the optic lobes with age

We observe higher levels of expression of DNA damage-associated genes in the optic lobes than in other parts of the brain (*Figure 5A*). We also see higher levels of DNA damage foci in the optic lobes than the central brain. We observe this signature even at 7 days in the OL, but it becomes stronger by 21 days (*Figure 5—figure supplement 1*). This is consistent with other studies that report that signatures of ageing appear gradually over the course of an organism's lifespan and not abruptly at later chronological ages (*Ben-Zvi et al., 2009*; *Labbadia and Morimoto, 2014*; *Shavlakadze et al., 2019*). We do not know whether the increased DNA damage signature we observe in the optic lobes is because the optic lobes intrinsically sustain higher levels of DNA damage or whether other parts of the brain are better equipped to resolve DNA lesions. We also see an upregulation of cell cycle-associated genes specifically in the optic lobes with age. The transcription of cell cycle genes and genes involved in the DNA damage response and repair are intimately coordinated and can be controlled by intersecting pathways. (*Chen et al., 2010*; *Herrup et al., 2013*; *Uxa et al., 2019*). Homology-directed repair of DNA lesions occurs in S and G2 phases of the cell cycle in actively dividing cells (*Herrup and Yang, 2007*). In other phases of the cell cycle, and after cell cycle exit, cells have to rely on error-prone non-homologous end joining mediated repair. It is tempting to speculate that re-entering the cell cycle allows postmitotic cells to repair DNA damage better and survive.

## Is polyploidy protective?

We and others observe a steady decline in the number of cells in the adult *Drosophila* brain with age (*Figure 6—figure supplement 1*; *Bates et al., 2019*; *Foo et al., 2017*). The continual loss of cells in the ageing brain may be analogous to wounding, which induces polyploidization or compensatory cellular hypertrophy in other *Drosophila* tissues (*Bretscher and Fox, 2016*; *Cohen et al., 2018*; *Losick et al., 2013*), (*Tamori and Deng, 2013*). We suggest neurons and glia in the adult brain may employ a similar strategy, to compensate for cell loss in a non-autonomous fashion. When we induce damage that does not increase cell death in young brains (*Figure 6B,D*) we observe an earlier increase in polyploidy, suggesting that in younger animals polyploidy can be an adaptive response to damage. By contrast in older animals, we find that polyploidy can protect from acute cell loss. However, levels of subsequent polyploidy do not further increase in aged animals, suggesting there is a permissive window for damage-induced polyploidy during adulthood (*Figure 6*). This may explain why levels of polyploidy plateau after 3–4 weeks of age in various strains (*Figure 1*). It will be interesting to further test the nature of this compensation for cell loss in early adulthood by performing genetic experiments to ablate specific cell types or sub-populations of cells.

## How does polyploidy relate to neurodegeneration?

Over the past two decades, several studies have reported an interesting correlation between neurodegeneration and cell cycle re-entry in neurons (*Chen et al., 2010*; *Frade and Ovejero-Benito, 2015*; *Herrup, 2012*; *Moh et al., 2011*; *Rimkus et al., 2008*; *Yang and Herrup, 2005*). Most of these observations are from post-mortem brains containing neurons expressing cell cycle genes or exhibiting hyperploidy (>2N DNA content). More hyperploidy is observed in brains of patients with preclinical Alzheimer's compared to age-matched controls, which has led to the hypothesis that cell cycle re-entry may precede cell death and neurodegeneration. Whether cell cycle re-entry is a cause or a consequence of neurodegeneration has been difficult to test, since both are associated with age and damage. Our data suggest that re-entry into the cell cycle may be a normal physiological response to the accumulation of damage in early adulthood and that it can serve a beneficial and protective function in neurons and glia. However, we do not know how polyploidy may impact neuronal and glial function and whether it may become detrimental over time. In geriatric animals (beyond 4 weeks) we observe increased variation in the levels of polyploidy and we note that a subset of animals also exhibit extreme levels of cell death (*Figure 6C*). It is possible that these animals represent a fraction of the aged population that exhibit neurodegeneration. Our single-animal assays will be essential to identify these outliers for further study.

# Materials and methods

## Key resources table

| Reagent type (species) or resource | Designation | Source or reference | Identifiers | Additional information |
|---|---|---|---|---|
| Genetic reagent (*D. melanogaster*) | *w*[1118] | Bloomington *Drosophila* Stock Center | BDSC 5905 | isogenic |
| Genetic reagent (*D. melanogaster*) | *Canton-S* | O. Shafer lab | n/a | WT |
| Genetic reagent (*D. melanogaster*) | *Oregon-R* | C. Collins lab | n/a | WT |
| Genetic reagent (*D. americana*) | *Drosophila americana* | P. Wittkopp lab | n/a | Non melanogaster *Drosophila* |
| Genetic reagent (*D. mauritiana*) | *Drosophila mauritiana* | P. Wittkopp lab | n/a | Non melanogaster *Drosophila* |
| Genetic reagent (*D. melanogaster*) | *glass*[60J] | O. Shafer lab | n/a | Mutant for glass |
| Genetic reagent (*D. melanogaster*) | *w;nSyb-GAL4/Cyo* | M. Dus lab | n/a | pan-neuronal |
| Genetic reagent (*D. melanogaster*) | *w;+;nSyb-GAL4* | M. Dus lab | n/a | pan-neuronal |
| Genetic reagent (*D. melanogaster*) | *w;UAS-nGFP; Repo-GAL4, tubulin GAL80TS* | Buttitta lab stocks | n/a | pan-glial |
| Genetic reagent (*D. melanogaster*) | *w;UAS-nGFP* | Buttitta lab stocks | n/a | UAS nuclear GFP |
| Genetic reagent (*D. melanogaster*) | *w;+;UAS-nGFP* | Buttitta lab stocks | n/a | UAS nuclear GFP |
| Genetic reagent (*D. melanogaster*) | *w;Moody-GAL4* | C. Collins lab via Klambt Lab | n/a | Sub-perineurial glia |
| Genetic reagent (*D. melanogaster*) | *y,w;mz19-mCD8::GFP* | Bloomington *Drosophila* Stock Center | BDSC 23300 | Antennal lobe projection neuron |
| Genetic reagent (*D. melanogaster*) | *w*[1118]*;ELAV-GAL4,UAS-nGFP* | Bloomington *Drosophila* Stock Center | BDSC 49226 | pan-neuronal |
| Genetic reagent (*D. melanogaster*) | *y,w;breathless-GAL4* | DGRC Kyoto | 105276 | Trachea |
| Genetic reagent (*D. melanogaster*) | *w-;GAD1-GAL4/SM6* | O. Shafer lab | n/a | GABAergic |
| Genetic reagent (*D. melanogaster*) | *w-;OK371-GAL4, UASn-GFP* | Buttitta lab stocks | n/a | Glutamatergic |
| Genetic reagent (*D. melanogaster*) | *w;ChaT-GAL4* | O. Shafer lab | n/a | Cholinergic |
| Genetic reagent (*D. melanogaster*) | *w*[1118]*;+;GMR-12C11-GAL4* | Bloomington *Drosophila* Stock Center | BDSC 76324 | Tm3a |
| Genetic reagent (*D. melanogaster*) | *w*[1118]*;+;GMR-42H01-GAL4* | Bloomington *Drosophila* Stock Center | BDSC 48150 | Dm9 |
| Genetic reagent (*D. melanogaster*) | *w*[1118]*;+;GMR-23G11-GAL4* | Bloomington *Drosophila* Stock Center | BDSC 49043 | Dm4 |

*Continued on next page*

*Continued*

| Reagent type (species) or resource | Designation | Source or reference | Identifiers | Additional information |
|---|---|---|---|---|
| Genetic reagent (*D. melanogaster*) | $w^{1118};+;GMR$-30B06-GAL4 | Bloomington *Drosophila* Stock Center | BDSC 47529 | Dm10 |
| Genetic reagent (*D. melanogaster*) | $w^{1118};+;GMR$-26H07-GAL4 | Bloomington *Drosophila* Stock Center | BDSC 49204 | Dm2 |
| Genetic reagent (*D. melanogaster*) | *y;w;NP3233-GAL4/Cyo* | DGRC Kyoto | 113173 | Astrocyte-like |
| Genetic reagent (*D. melanogaster*) | *y;w;NP2222-GAL4/Cyo* | DGRC Kyoto | 112830 | Cortex glia |
| Genetic reagent (*D. melanogaster*) | *w;mz97-GAL4* | C. Collins lab via Klambt Lab | n/a | Wrapping glia |
| Genetic reagent (*D. melanogaster*) | *y,w,UAS-mCD8::RFP,LexA$_{op}$2-mCD8::GFP; CoinFLP-LexA::GAD.GAL4* | Bloomington *Drosophila* Stock Center | BDSC 59270 and 59271 | CoinFLP |
| Genetic reagent (*D. melanogaster*) | *y,w,hs-FLP; LexA$_{op}$-RFP$_{nls}$; UAS-GFP$_{nls}$* | Buttitta lab stocks | n/a | *hs-FLP* used with with CoinFLP nuclear GFP and RFP |
| Genetic reagent (*D. melanogaster*) | *ey-FLP* | Bloomington *Drosophila* Stock Center | BDSC 5576 | ey-FLP |
| Genetic reagent (*D. melanogaster*) | *y,sev,w; UAS-cdc6$^{RNAi}$* | Bloomington *Drosophila* Stock Center | BDSC 55734 | cdc6KD |
| Genetic reagent (*D. melanogaster*) | *w; UAS-geminin$^{RNAi}$* | Bloomington *Drosophila* Stock Center | BDSC 30929 and 50720 | gemininKD |
| Genetic reagent (*D. melanogaster*) | *w1118; GUS-p53* | Bloomington *Drosophila* Stock Center | BDSC 6584 | UAS-p53WT |
| Genetic reagent (*D. melanogaster*) | *y,w1118; UAS-p53 $^{259N}$* | Bloomington *Drosophila* Stock Center | BDSC 6582 | UAS-p53DN |
| Antibody | anti-ELAV (rat monoclonal) | Developmental Studies Hybridoma Bank | Rat-ELAV-7E8A10 | 1: 100 |
| Antibody | anti-pH2AV (mouse monoclonal) | Developmental Studies Hybridoma Bank | UNC93-5.2.1 | 1: 100 |
| Antibody | anti-Repo (mouse monoclonal) | Developmental Studies Hybridoma Bank | 8D12 | 1: 100 |
| Antibody | anti-Lamin (mouse monoclonal) | Developmental Studies Hybridoma Bank | ADL67.10 | 1: 100 |
| Antibody | Alexa Fluor 568 anti-mouse (goat polyclonal) | ThermoFisher | A11031 | 1: 1000 |
| Antibody | Alexa Fluor 568 anti-rat (goat polyclonal) | ThermoFisher | A11077 | 1: 1000 |

*Continued on next page*

Continued

| Reagent type (species) or resource | Designation | Source or reference | Identifiers | Additional information |
|---|---|---|---|---|
| Antibody | Alexa Fluor 488 anti-mouse (goat polyclonal) | ThermoFisher | A11029 | 1: 1000 |
| Antibody | Alexa Fluor 488 anti-rat (donkey polyclonal) | ThermoFisher | A21208 | 1: 1000 |
| Other | DAPI | Sigma-Aldrich | D9542 | 1: 1000 |
| Other | Dye-cycle violet | ThermoFisher | V35003 | 2: 1000 |
| Other | Sytox Green | ThermoFisher | S7020 | 2: 1000 |
| Other | Propidium Iodide | Sigma-Aldrich | P4170 | 2.25: 1000 |

## Fixation, Immunostaining and Imaging

*Drosophila* brains were dissected in 1X Phosphate buffered saline (PBS) and fixed in 4% Paraformaldehyde (PFA) in 1X PBS for 25 min. Tissues were permeabilized in 1X PBS+0.5% Triton-X, blocked in 1X PBS, 1% BSA 0.1% Triton-X. (PAT) Antibody staining was performed at specified concentrations in PAT (*Supplementary file 1*) overnight at 4°C, washed, blocked in PBT-X (1X PBS, 2% Goat serum 0.3% Triton-X) prior to incubation with secondary antibody either for 4 hr at RT or overnight at 4°C. DAPI staining was performed after washes, brains were wet-mounted in vectashield H1000. All imaging was performed on either a Leica SP5 or SP8 laser scanning confocal microscopes. For EdU incorporation assays, flies were placed on 10 mM EdU containing food with food coloring for 3 days prior to dissection. Only flies with visibly colored abdomens were dissected. Click-iT Plus staining with picolyl Azide was done as per the protocol recommended by ThermoFisher.

## Fly husbandry

Flies were reared and aged in a protocol modified from *Linford et al., 2013*. Ageing flies were collected soon after eclosion as virgin males and females and segregated into vials containing no more than 20 flies/vial. Ageing flies were flipped onto fresh Bloomington Cornmeal food every 5–7 days. A list of all fly stocks used in this study is supplied as a table in *Supplementary file 2*.

## Image quantification

For pH2AV quantification, five non-overlapping Regions of Interest (ROIs) were chosen per brain region per brain. Average Intensity of pH2AV and DAPI per ROI were computed on individual channels using ImageJ. All brains were imaged at the same laser intensity and gain settings at different ages.

CoinFLP double-labeled cell counting was performed manually. Individual optic lobes were imaged at 100x magnification with 0.5 micron Z-sections. Quantification was performed by cropping 2–5 confocal Z-sections at a time, performing maximum intensity projections of each cropped image, and counting cells that showed DAPI, GFP and RFP signal overlap. Lamin staining was used to discern nuclear boundaries for DAPI Integrated Intensity measurements of 132 cells using FIJI. DAPI intensity was normalized to diploid cells (2N) measured on the same slide.

## Heat shock protocol

CoinFLP labeling was induced with heat shock induction. Flies were placed in plastic vials and completely submerged in 37°C water bath for 15 min. For 'early-FLP', flies were moved back to 23°C and dissected at day 1 or day 10. For 'late-FLP', heat shock induction was performed 24 hr prior to dissection. All incubations and culturing except heat shock was performed at 23°C. We noted that the frequency of CoinFLP flipping resulted in a ratio of LexA:GAL4 expressing cells that is between 4:1 and 4.6:1 (*Bosch et al., 2015*). We calculated the expected number of double labeled cells for 'late-FLP' in *Figure 4* and *Figure 4—figure supplement 2* as follows: If flipping is complete (100%

of cells flip), we would expect 80% of diploid cells to label red and ~20% to label green. Only 4% of all polyploid cells will label green/green (probability of green is 0.2 therefore 0.2* 0.2 = 0.04*100), 64% red/red (probability of red is 0.8 thus, 0.8*0.8 = 0.64*100) and 32% will label green/red or red/green and appear yellow or 'double-labeled'. If we assume that about 20% of cells in the optic lobes are tetraploid (20% of ~30,000 = 6,000 cells), we can expect 1,920 cells (32% of 6,000) to label yellow per optic lobe under 100% flipping conditions. If we flip ~50% of cells, we expect about 900 yellow cells per optic lobe. In our measurements the amount of flipping was variable from animal to animal and we estimate that in our samples with the lowest flipping we flip about ⅓ of cells and with our strongest flipping we label about ¾ cells.

## Flow cytometry

Fly brains were dissected in PBS and transferred to 1.5 mL microcentrifuge tube caps containing 100 uL of solution containing 9:1 Trypsin-EDTA: 10XPBS with 1 µL Dyecycle Violet and/or 1.12 µL PI or 1 µL Sytox green. Brains were incubated for 20 min in the microcentrifuge tube caps, triturated using low retention p200 pipette tips for 60 s then transferred into the microcentrifuge tubes containing 400 µL of the trypsin-EDTA solution with dyes and capped, and incubated further for 45 min at room temperature without agitation. After incubation, each sample was diluted with 500 µL 1XPBS and gently vortexed at speed eight before being loaded onto Attune or Attune NxT flow cytometer for flow cytometry analysis. The Attune had a laser configuration of a violet laser (VL,405nm) with six bandpass (BP) filters and a blue laser (BL,488nm) with three bandpass filters. The Attune NxT is configured with VL (6 BP filters), BL with 2 BP filters, a yellow laser (YL, 561 nm) with 3 BP filters and a red laser (RL, 637 nm) with 3 BP filters. The detection of DyeCycle Violet was performed using VL1 (Emission filter 450/40), GFP and Sytox Green using BL1 (Emission filter 530/30), RFP and PI using BL2 (Emission filter 574/24) on the Attune and YL1 (585/16) on the Attune NxT. A flow rate of 100 to 500 µl/second was used for sample acquisition and a minimum of 20,0000 events gated as 'non doublets' (*Figure 1—figure supplement 1*) were acquired per sample. Gating Strategy is graphed in *Figure 1—figure supplement 1*. Briefly, all cells were plotted on forward vs side scatter (FSC vs. SSC), gated to eliminate debris. Subsequently, 'non-debris' were plotted on VL1(DNA) vs FSC and gated to eliminate unstained events. A third gate was applied plotting VL1(DNA)-H vs VL1(DNA)-A (voltage pulse area vs.height) to eliminate doublets. All events in gate three were further subjected to GFP/DNA/PI content analysis.

## RNA sequencing

10 CNSs from *Canton-S* females and males raised at 25℃ on Cornmeal/Dextrose food under normal 12 h L/D cycles at ZT = 2, were dissected into optic lobes, VNC and central brain at the indicated ages with three biological replicates for each sex, age and region (72 samples total). Tissues were directly dissolved into TRIZOL-LS. (Invitrogen) and RNA was prepared as directed by the manufacturer. Total RNA (2–5 µg) was provided to the University of Michigan Sequencing Core for polyA selection and unstranded mRNA library preparation for the Illumina HiSeq4000 platform.

## RNAseq data analysis and GO term analysis

RNAseq analysis was performed at the U.Michigan Bioinformatics Core using the following pipeline:
   1.Read files from the Sequencing Core were concatenated into single fastq files for each sample. 2. Quality of the raw reads data for each sample was checked using FastQC(version v0.11.7). 3. Adaptors and poor quality bases were trimmed from reads using bbduk from the BBTools suite (v37.90).4. Quality processed reads were aligned to the Ensembl Dm6 genome using STAR (v2.6.1a) with quantMode GeneCounts flag option set to produce gene level counts. MultiQC (v1.6a0) was run to summarize QC information for raw reads, QC processed reads, alignment, and gene count information. Differential expression analyses were carried out using DESeq2 (v1.14.1). Data were pre-filtered to remove genes with 0 counts in all samples. Normalization and differential expression was performed with DESeq2, using a negative binomial generalized linear model. Plots were generated using variations or alternative representations of native DESeq2 plotting functions, ggplot2, plotly, and other packages within the R environment.
   Genes called as at least 2-fold differentially expressed between day 2 and day 21 were examined for enriched GO terms using target and background unranked lists in GOrilla and redundant GO

terms were filtered using ReviGO. GO term Enrichment is presented as the -log10 of the p-value with a cutoff at p-values higher than 10$^{-3}$. The full dataset has been uploaded to GEO and can be found using the accession number: GSE153165.

## Acknowledgements

We would like to thank members of the Buttitta and Cheng-Yu Lee labs for helpful discussions and suggestions. We also thank Yiqin Ma, Kerry Flegel, Chelsea Yu, Emily Lerner, Ashley Francis, Andrew White and Emily Rozich for help with experiments. We would also like to thank the Neuroscience and *Drosophila* research communities at UM for their support for providing fly stocks, particularly the labs of Cheng-Yu Lee, Monica Dus, Catherine Collins, Josie Clowney, Trisha Wittkopp, Scott Pletcher and Orie Shafer. We acknowledge support from the Bioinformatics Core and Chris Sifuentes of the University of Michigan Medical School's Biomedical Research Core Facilities. This study was funded by NIH R21 AG047931, NIH R01 GM127367, ACS Scholar Award RSG-15-161-01-DDC. SN was supported in part by the Barbour Scholarship (University of Michigan).

## Additional information

### Funding

| Funder | Grant reference number | Author |
|---|---|---|
| National Institutes of Health | AG047931 | Laura A Buttitta |
| National Institutes of Health | GM127367 | Laura A Buttitta |
| American Cancer Society | RSG-15-161-01-DDC | Laura A Buttitta |
| University of Michigan | Barbour Scholarship | Shyama Nandakumar |

The funders had no role in study design, data collection and interpretation, or the decision to submit the work for publication.

### Author contributions

Shyama Nandakumar, Conceptualization, Data curation, Formal analysis, Investigation, Writing - original draft, Writing - review and editing; Olga Grushko, Conceptualization, Investigation, Writing - review and editing; Laura A Buttitta, Conceptualization, Resources, Formal analysis, Supervision, Funding acquisition, Investigation, Writing - original draft, Project administration, Writing - review and editing

### Author ORCIDs

Shyama Nandakumar (D) https://orcid.org/0000-0003-0624-3452
Laura A Buttitta (D) https://orcid.org/0000-0002-5064-0650

### Decision letter and Author response

Decision letter https://doi.org/10.7554/eLife.54385.sa1
Author response https://doi.org/10.7554/eLife.54385.sa2

## Additional files

### Supplementary files

- Supplementary file 1. A list of antibodies and stains used.
- Supplementary file 2. A list of *Drosophila* stocks used.
- Supplementary file 3. Proportions of cell types polyploid in the whole brain.
- Supplementary file 4. Proportions of polyploid cell types in the OL.
- Transparent reporting form

## Data availability

Sequencing data have been deposited in GEO under accession code GSE153165.

The following dataset was generated:

| Author(s) | Year | Dataset title | Dataset URL | Database and Identifier |
|---|---|---|---|---|
| Nandakumar S, Buttitta L | 2020 | Polyploidy in the adult Drosophila melanogaster brain | https://www.ncbi.nlm.nih.gov/geo/query/acc.cgi?acc=GSE153165 | NCBI Gene Expression Omnibus, GSE153165 |

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
