## [Decision Letter]

**Acceptance summary:**

Polyploidization is hypothesized to promote survival in response to the accumulation of age-related DNA damage. Polyploid and aneuploid cells in the brain have been described before, as is their accumulation during ageing and disease. However, the idea of this being a response to DNA damage is novel. The authors make an effort to associate ploidy changes in adult animals with brain regions and cell types, which is an important unaddressed question in the field. The experiments aiming to demonstrate that polyploid cells in the brain are a consequence of cell cycle re-entry are also of interest.

**Decision letter after peer review:**

Thank you for submitting your article "Polyploidy in the adult *Drosophila* brain" for consideration by *eLife*. Your article has been reviewed by three peer reviewers, and the evaluation has been overseen by K VijayRaghavan as the Senior and Reviewing Editor. The following individual involved in review of your submission has agreed to reveal their identity: Grace E Boekhoff-Falk (Reviewer #3).

The reviewers have discussed the reviews with one another and the Reviewing Editor has drafted this decision to help you prepare a revised submission.

Summary:

The manuscript entitled "Polyploidy in the adult *Drosophila* brain" by Nandakumar and colleagues describes the age-related accumulation of polyploid cells in the *Drosophila* brain and proposes this mechanism as a response to DNA damage. Polyploidization is hypothesized to promote survival in response to the accumulation of age-related DNA damage.

Polyploid and aneuploid cells in the brain have been described before, as is their accumulation during aging and disease. However, the idea of this being a response to DNA damage is novel. The authors make an effort to associate ploidy changes in adult animals with brain regions and cell types, which is an important unaddressed question in the field. The experiments aiming to demonstrate that polyploid cells in the brain are a consequence of cell cycle reentry are also of interest. Overall, while the study is of interest, it lacks quantitative follow-up experiments and includes several correlative studies as well as overstatements of many conclusions. These and other concerns are outlined below. The reviews also involve suggestions for experiments. The authors should decide which are best suited to speedily and robustly address the requirements for clear evidence for their assertions, or where discussion of previous work from their labs will suffice. In any event, the experiments are doable in a 2-month period. The authors should make every effort to firmly address the reviewers' concerns so that a decision can be taken by avoiding further revisions. The work will represent a strong contribution to the field if the substantive concurs are well-addressed.

Essential revisions:

1) The study is entirely based on ploidy detection by FACS using a custom assay to measure DNA content; however, this assay has not been benchmarked to determine its sensitivity and specificity. The only validation is correlative with the frequency of EdU positive cells shown in Figure 1—figure supplement 1F. The FACS based method needs to be validated with an independent technique (i.e. FISH, single cell sequencing) in order to strengthen the following statement "We were therefore surprised to find a distinct population of cells with DNA content of 4C and up to >16C appearing in brains of aged animals of various genotypes under normal culture conditions (Figure 1—figure supplement 1G)". We recommend that polyploidy be also shown in a second way such as high resolution imaging and single cell fluorescence measurements of DNA, or using DNA FISH probes.

2) The animals analyzed in this study are 1 week increment up to 8 weeks; the age related increased in polyploidy begins at 7 days and continues to accumulate up to 21 days (week 3) after which reaches a plateau. This age group should be considered middle age and therefore the data do not support the statement in the Abstract "that "polyploidy protects against DNA damage-induced cell death".

3) The COINFLP system applied to demonstrate that cells reenter the cell cycle shows correlative data; green, red and yellow cells should be FACS enriched to measure the frequency of polyploid cells in the yellow fraction. This is important because while it is understood that yellow labelled cells are assumed to be polyploid this has not been formally demonstrated. Likewise, in the silencing experiments (cdc6 and Geminin) yellow cells are assumed polyploid and even labelled as such in Figure 4G but no quantification of DNA content is ever provided.

4) The RNA seq data have not been validated, and again show correlative results.

5). There are three main part to the model presented.

a) DNA damage accumulating with age. This result is not surprising and there are several examples that show this in various tissues, including in brains. We also think that the staining and imaging for pH2AV must be greatly improved. There are several other experiments that could benefit from images, or better images. The authors must look at their experiments and see where this can be added meaningfully or experiments done.

b) "Exogenous" DNA damage leading to increased polyploidy. I think the authors were very careful to use the exogenous damage here as their analysis of ploidy after PQ treatment might not reflect normal polyploidy due to normal levels of stress and age. In addition, their analysis is done in bulk tissue and not on a single cell basis to correlate a cell's DNA damage with its ploidy. It would be interesting to further test the idea that DNA damage induces polyploidy by over-expressing or knocking down a DNA repair factor such as Prp19 and then measuring the proportion of cells that are polyploid. Again, this is experimentally feasible speedily, but if the authors choose not to do this, these must have a cogent and strong reasoning.

c) Polyploidy protecting from cell death. We think this idea is founded in the literature from other cell types, and thus it is a reasonable hypothesis in the fly brain. However, we do not think the experiment presented is sufficient to make this a solid conclusion. A change in the percent of dying cells from 7 to 2.5 (diploid vs polyploid) from cell counts that are already very small and from brains that are pooled does not seem like a reliable experiment. We think additional experiments should be done. For example, the authors could use a GAL4 expressed in a restricted sub-population of cells (ex. SPGs via R54C07-GAL4 which was used in Kremer et al., 2017) to label and directly count the number of cells in the optic lobes via light microscopy (it should be <100 cells per lobe). Then, RNAi against cdc6 and geminin (as in Figure 4G) could be used to manipulate the proportion of cells that are polyploid. Using such an approach would allow the authors to link the proportion of cells that are polyploid to the total number of cells and thus directly show that polyploidy protects the cells from DNA damage-induced cell death.

6) A criticism of the work as presented is that the vast majority of the polyploid cells are tetraploid, yet the authors do not address the possibility that these cells simply have replicated their DNA prior to dividing (i.e. are transiently polyploid) as versus remaining indefinitely polyploid. Earlier work from other laboratories suggests that this unlikely, and the manuscript would be improved by a discussion of the possibilities.

7) The authors state that up to 20% of the cells in the optic lobes are polyploid. However, later in the text (and the figure legends) the number is 25%. Please be consistent.

8) In the Discussion, the authors propose that polyploidization may permit cells to maintain contacts when some of their neighbors die. This hypothesis also was proposed by Unhavaithaya and Orr-Weaver who should be cited. The BBB example is particularly compelling b/c the contacts create the barrier.

9) The authors state “It is also possible that we see higher levels of DNA damage foci in the optic lobes because they have more polyploid cells” but do not offer an explanation for why this could be biologically relevant.

10) Also, the authors state “We see an upregulation of cell cycle-associated genes specifically in the optic lobes with age. The transcription of cell cycle genes and genes involved in the DNA damage response and repair are intimately coordinated and can be controlled by the same factors.” Please complete the argument – namely that upregulation of DNA damage response and cell cycle genes may share an upstream regulatory mechanism.

11) The authors state “Whether cell cycle re-entry is a cause or a consequence of neurodegeneration has been difficult to test since both are associated with age and damage.” An earlier paper would seem to be relevant and could be cited: Mutations in String/CDC25 inhibit cell cycle re-entry and neurodegeneration in a *Drosophila* model of Ataxia-telangiectasia. Rimkus et al., 2008.

12) In Figure 1, there is an intriguing dip in the % polyploidy after day 21 in males and day 28 in females. What are the authors' thoughts about this? Could the initial wave of polyploidization result in death and then be followed by a wave of polyploidization that results in persistent, stress-resistant cells?

13) In Figure 3, the # of double-labeled cells from early flp significantly increases from Day 1 to Day 14 – doesn't this suggest there is some cell fusion? Also, image B' is from a 30 day animal, while the quantitation in C is from 14-day animals. What does the quantitation look like at 30 days? Does the number of yellow cells increase further?

14) For the data presented in Figure 4, the labeling of 300 cells/optic lobe is impressive. However, if there are ~20,000 cells/optic lobe, then the 300 cells would be only ~1-2% of the total, not the estimated 25% that you measure as >4N. Can you explain the discrepancy? (BTW, we think lineage tracing methods such as Permatwin under-report dividing cells in the adult brain).

15) For Figure 6, it would be interesting to pulse the flies with paraquat, then chase a few days without paraquat before assessing ploidy. What if the polyploid cells are en route to dying? This seems unlikely for UV irradiated flies, b/c the irradiation was done 5 days prior to dissection. However, in Figure 5D, the fact that the control and paraquat-treated lines cross at 21 days seems to support the idea that some of the polyploid cells in the paraquat-treated animals die and/or divide. Please comment.

16) In Figure 6—figure supplement 1. It appears that pH2AV levels are elevated in all neurons in the paraquat treated brains. However, only a fraction of these die. Why might this be?

---

## [Author Response]

Essential revisions:1) The study is entirely based on ploidy detection by FACS using a custom assay to measure DNA content; however, this assay has not been benchmarked to determine its sensitivity and specificity. The only validation is correlative with the frequency of EdU positive cells shown in Figure 1—figure supplement 1F. The FACS based method needs to be validated with an independent technique (i.e. FISH, single cell sequencing) in order to strengthen the following statement "We were therefore surprised to find a distinct population of cells with DNA content of 4C and up to >16C appearing in brains of aged animals of various genotypes under normal culture conditions (Figure 1—figure supplement 1G)". We recommend that polyploidy be also shown in a second way such as high resolution imaging and single cell fluorescence measurements of DNA, or using DNA FISH probes.

We agree this is an important issue.

Several years ago we attempted to resolve chromosome copy numbers using FISH on the *Drosophila* brain. We found this to be impossible using high resolution confocal microscopy, due to constitutive somatic homologue pairing (Joyce et al., 2012; McKee, 2004; Metz, 1916; Stevens, 1908). We therefore attempted to resolve homologues using STORM super-resolution microscopy and large FISH probes (~10kb). However we could not image deeper than the most superficial layer of cells using STORM. Despite several months of work, we were ultimately unable to apply FISH to conclusively resolve chromosome copy number in adult brains.

Image-based quantification of Dapi nuclear intensity is another common way to measure ploidy in tissues, but this is most accurate in flat epithelial tissues, where individual nuclei can be resolved and segmented for intensity measurements. This was extremely challenging in the adult brain, where many cells are closely overlapping in all three dimensions. However we agree confirming polyploidy with another method is an important issue, so we revisited this approach. This time we used nuclear lamin staining and nuclear Coin-FLP labeling to aid in manually resolving individual nuclei for Dapi measurements. In this revision, we present new high magnification confocal images of optic lobes from CoinFLP “late-FLP” animals where we induced labelling 24h prior to dissection and labeling with anti-lamin and Dapi. We used the lamin staining to mark nuclear boundaries and Dapi intensity to measure ploidy (normalized integrated intensity). We manually measured over 100 cells and found that the CoinFLP double positive or “yellow” cells always have a greater than diploid DNA content. This also agrees with new FACS data we have added on CoinFLP cells to address point #3. However it should be clarified that due to the stochastic nature of CoinFLP labeling, polyploid cells can also be single-labeled (e.g. green/green or red/red) or unlabeled (did not flip), which are also included in our ploidy measurements (Described in detail in point #14). This new data can be found in Figure 4—figure supplements 1 and 2.

2) The animals analyzed in this study are 1 week increment up to 8 weeks; the age related increased in polyploidy begins at 7 days and continues to accumulate up to 21 days (week 3) after which reaches a plateau. This age group should be considered middle age and therefore the data do not support the statement in the Abstract "that "polyploidy protects against DNA damage-induced cell death".

We thank the reviewers for this comment. We do agree that 7day animals cannot be considered “old” and we have changed the Abstract to read “in the adult brain” instead of “in the ageing brain”. We have also made a much clearer distinction in this revision about polyploidy in the early adult vs. aged brains (see Results section, Figure 6 and Discussion). Our RNA sequencing shows that 7day old optic lobes already exhibit a DNA damage-associated gene expression signature (Figure 5—figure supplement 1), suggesting that even early adult animals are coping with accumulating DNA damage and this is an issue we discuss further in this revision. This is similar to defects in proteostasis associated with aging that are already detectable during early adulthood, as described for *C. elegans* (Morimoto, 2020)

3) The COINFLP system applied to demonstrate that cells reenter the cell cycle shows correlative data; green, red and yellow cells should be FACS enriched to measure the frequency of polyploid cells in the yellow fraction. This is important because while it is understood that yellow labelled cells are assumed to be polyploid this has not been formally demonstrated. Likewise, in the silencing experiments (cdc6 and Geminin) yellow cells are assumed polyploid and even labelled as such in Figure 4G but no quantification of DNA content is ever provided.

We agree that showing that the CoinFLP “yellow” cells are polyploid via FACS will greatly strengthen our study. To address this we have performed additional FACS and have added a supplemental figure showing >97% of all CoinFLP double-labelled cells display >2C DNA content (Figure 4—figure supplement 2). We also provide verification of these FACS results with direct DAPI intensity measurements from confocal images (please see response to point #1 and Figure 4—figure supplement 1).

With regard to the knockdown experiments, we apologise for the lack of clarity. The changes in ploidy for the cdc6 and geminin experiments were measured by flow cytometry, not CoinFLP. We knocked down the respective genes in neurons (which are ~90% of cells in the brain) using the driver nSybGAL4 and measured changes in the % of cells showing >2C DNA content by flow cytometry. (Figure 4G) We have clarified this in the figure legend as well as the text.

4) The RNA seq data have not been validated, and again show correlative results.

We thank the reviewer for encouraging us to provide additional information on the RNAseq dataset to strengthen the paper. There are two commonly used types of validation, 1. Comparison to an external dataset to verify sample quality/comparisons and 2. Individual gene fold-change confirmation by another method – most commonly qRT-PCR. Over the past 3 years we have anecdotally found that qRT-PCR is actually less reproducible and accurate for measuring gene expression across 3-6 biological replicates than RNAseq (e.g. data within (Flegel et al., 2016) ). This has been explored more thoroughly by others, which have also come to the conclusion that RNAseq with 3 or more replicates is more reproducible and accurate to measure gene expression change than qRT-PCR, especially when no accurate, unchanged reference gene is available for normalization, as is the case for brain aging. (See for examples: (Griffith et al., 2010;Pombo et al., 2017;Zhang et al., 2019)) We therefore focused on ensuring the accuracy of our comparisons by using external datasets to confirm that our RNAseq correctly identifies previously published age-specific gene expression changes, region-specific expression and sex-specific gene expression patterns. See new Figure 5—figure supplement 2 and Supplementary file 3 and references therein.

5) There are three main part to the model presented.a) DNA damage accumulating with age. This result is not surprising and there are several examples that show this in various tissues, including in brains. We also think that the staining and imaging for pH2AV must be greatly improved. There are several other experiments that could benefit from images, or better images. The authors must look at their experiments and see where this can be added meaningfully or experiments done.

We have included better representative images of pH2AV in Figure 5C through D’. We have also included more and higher resolution images of CoinFLP labeled cells in Figure 4—figure supplement 1. Please note that ELAV immmuno-fluorescence in older brains does not look as strongly nuclear as in young brains and appears as subnuclear puncta. This has also been observed in work from other labs studying the fly brain at ages beyond 1 week. (Frost et al., 2016).

b) "Exogenous" DNA damage leading to increased polyploidy. I think the authors were very careful to use the exogenous damage here as their analysis of ploidy after PQ treatment might not reflect normal polyploidy due to normal levels of stress and age. In addition, their analysis is done in bulk tissue and not on a single cell basis to correlate a cell's DNA damage with its ploidy. It would be interesting to further test the idea that DNA damage induces polyploidy by over-expressing or knocking down a DNA repair factor such as Prp19 and then measuring the proportion of cells that are polyploid. Again, this is experimentally feasible speedily, but if the authors choose not to do this, these must have a cogent and strong reasoning.

We agree with the reviewers’ comment and we are keen to perform additional experiments to address this. However we received the reviewers comments a few weeks before our research operations shut down due to COVID-19. Our lab has only partially re-opened under a part-time shift schedule as of July 1st and we have finally acquired several lines to attempt to address this issue. These lines include UAS-Prp19 from Dr.Thomas Flatt’s lab, and other lines from various stock centers to knock down key players in the DNA damage response that show changes with age such as *Irbp, Ku80, xrp1, rad51b, lig3, mus308* and *xpd*. We plan to knock these factors down in neurons as well as glia to interrogate which DNA damage pathways may play a role in inducing polyploidy. Since these experiments require 1-3 week age adult animals, we have written to the editor that a realistic timeline for these experiments is an additional 2-3 months. We would like to propose to perform these experiments and submit our findings as a preprint linked to this manuscript. We suspect DNA damage leads to increased polyploidy non-autonomously, however at this time we cannot rule out cell-autonomous effects. In this revision we also provide additional data addressing this issue (see response to point 5C and response to point #9).

c) Polyploidy protecting from cell death. We think this idea is founded in the literature from other cell types, and thus it is a reasonable hypothesis in the fly brain. However, we do not think the experiment presented is sufficient to make this a solid conclusion. A change in the percent of dying cells from 7 to 2.5 (diploid vs polyploid) from cell counts that are already very small and from brains that are pooled does not seem like a reliable experiment. We think additional experiments should be done. For example, the authors could use a GAL4 expressed in a restricted sub-population of cells (ex. SPGs via R54C07-GAL4 which was used in Kremer et al., 2017) to label and directly count the number of cells in the optic lobes via light microscopy (it should be <100 cells per lobe). Then, RNAi against cdc6 and geminin (as in Figure 4G) could be used to manipulate the proportion of cells that are polyploid. Using such an approach would allow the authors to link the proportion of cells that are polyploid to the total number of cells and thus directly show that polyploidy protects the cells from DNA damage-induced cell death.

We thank the reviewers for suggesting this experiment. Using a driver for SPG may not be ideal because the SPGs become polyploid very early in development, and their endocycles and endomitoses are critical for the growth of the brain in the larval stages (Unhavaithaya and Orr-Weaver, 2012). Since these cells will already be highly polyploid at eclosion, we will not be able to manipulate their ploidy specifically in the adult brain. We would instead like to manipulate the ploidy of a cell type that is diploid during development and becomes polyploid specifically in the adult, such as the glia of the optic chiasm. We have finally obtained lines to allow us to manipulate ploidy in these cells, and we propose to include our findings in a preprint linked to this manuscript.

We have also added new data that addresses this question in another way. To show that polyploid cells are protected from cell death induced by DNA damage, we performed an experiment similar to that shown in Figure 6G. We induced DNA damage in flies 21d old (a time when the OL shows high levels of polyploidy). We then performed a timecourse to identify the window where cells show high levels of death in response to UV. We see very high levels of cell death (>15% of cells incorporate PI) 24h post UV exposure. When we measured the ploidy of the dying cells, we see that while ~24% of diploid cells undergo cell death, only ~3% of the polyploid cells do. This much more convincingly shows that polyploid cells are protected from DNA damageinduced cell death. This data is now included in Figure 6.

With regards to pooling brains, we understand that it does not give us information about individual brains, but since the proportion of polyploid cells that are dying are so low, we needed to pool brains to ensure that our statistics are robust.

6) A criticism of the work as presented is that the vast majority of the polyploid cells are tetraploid, yet the authors do not address the possibility that these cells simply have replicated their DNA prior to dividing (i.e. are transiently polyploid) as versus remaining indefinitely polyploid. Earlier work from other laboratories suggests that this unlikely, and the manuscript would be improved by a discussion of the possibilities.

We have considered the possibility that the cells with 4C DNA content are simply in G2 and poised to undergo mitosis. However, we have stained for PH3 to look for mitosis in nearly 100 adult brains at different ages and have never observed a convincing mitotic event. We have also stained for G2 markers (e.g. CycA protein) many times at many ages and we have never resolved any convincing, reproducible staining. Importantly, most of the polyploid cells are neurons. We know from other experiments in the developing brain that pushing a neuron into mitosis is catastrophic and results in neuronal death. It is possible that a low level of mitoses occur that we have missed due to rapid cell death. We address this further in the revised Discussion.

7) The authors state that up to 20% of the cells in the optic lobes are polyploid. However, later in the text (and the figure legends) the number is 25%. Please be consistent.

Thank you to the reviewers for catching this. We have made the change in the text to be more consistent.

8) In the Discussion, the authors propose that polyploidization may permit cells to maintain contacts when some of their neighbors die. This hypothesis also was proposed by Unhavaithaya and Orr-Weaver who should be cited. The BBB example is particularly compelling b/c the contacts create the barrier.

Thank you for this helpful comment, we have included this in the revised text.

9) The authors state “It is also possible that we see higher levels of DNA damage foci in the optic lobes because they have more polyploid cells” but do not offer an explanation for why this could be biologically relevant.

We have removed this statement. During the original manuscript drafting we considered the possibility that polyploid cells accumulate DNA damage without dying, and therefore may over time, ultimately acquire more DBSs than diploid cells thereby biasing the pH2Av staining pattern. To test whether this is true, we measured pH2Av fluorescence and Dapi integrated intensity to examine the correlation of ploidy and number of DSBs. We found no correlation between ploidy and pH2Av staining (See Author response image 1), so we have removed this comment.

10) Also, the authors state “We see an upregulation of cell cycle-associated genes specifically in the optic lobes with age. The transcription of cell cycle genes and genes involved in the DNA damage response and repair are intimately coordinated and can be controlled by the same factors.” Please complete the argument – namely that upregulation of DNA damage response and cell cycle genes may share an upstream regulatory mechanism.

Thank you for this helpful comment, we have included this in the revised text.

11) The authors state “Whether cell cycle re-entry is a cause or a consequence of neurodegeneration has been difficult to test since both are associated with age and damage.” An earlier paper would seem to be relevant and could be cited: Mutations in String/CDC25 inhibit cell cycle re-entry and neurodegeneration in a Drosophila model of Ataxia-telangiectasia. Rimkus et al., 2008.

Thank you for pointing this out to us. We have included this citation in our Discussion.

12) In Figure 1, there is an intriguing dip in the % polyploidy after day 21 in males and day 28 in females. What are the authors' thoughts about this? Could the initial wave of polyploidization result in death and then be followed by a wave of polyploidization that results in persistent, stress-resistant cells?

This is an interesting point. We looked closely and we do not observe any significant difference in cell death between 21 and 28 days in males or 28 and 35 days in females (Figure 6). In both cases, most of the cells that die are diploid, so we do not have evidence for a wave of cell death eliminating polyploid cells. Although it is possible that it is transient and rapid, and therefore missed in our timecourse.

Another possibility is that there is a wave of mitosis, reducing tetraploid cells to diploid. We have examined this time window for mitoses using PH3 staining and have never observed any mitoses at this age. However, it is again possible that we have missed a transient wave of mitosis. If this is the case, we would expect higher ploidies (e.g. above tetraploid) to remain or increase, while tetraploid cells decrease. We examined the data more carefully to see if this occurs. When we look at male or female Canton-S after 28 days we see both tetraploid and >4N cells dip, suggesting the dip in polyploidy cannot be solely explained by a wave of mitosis resolving tetraploid cells (Figure 1D shows data for males). We do not have an explanation for this dip after 28 days, but a more detailed analysis of alternative modes of cell division or a finer timecourse analysis of cell death at these ages is needed.

13) In Figure 3, the # of double-labeled cells from early flp significantly increases from Day 1 to Day 14 – doesn't this suggest there is some cell fusion? Also, image B' is from a 30 day animal, while the quantitation in C is from 14-day animals. What does the quantitation look like at 30 days? Does the number of yellow cells increase further?

Thank you for pointing this out. Although we believe the majority of these yellow cells from the early flip are sub-perineurial glia we do not rule out a very low level of cell fusion. However, the level of fusion does not account for the vast majority of the polyploid cells we observe in later stages. We have changed our language to clarify this point.

The representative image in Figure 4B shows CoinFLP “late-FLP” labelled OLs from 30 days from the original CoinFLP lines which express membrane tagged UAS-RFP and LexA_op_-GFP. The membrane GFP and RFP posed two issues for quantifications; 1. We encountered membrane staining that was difficult to discern due to cell wrapping and 2. Like (Bosch et al., 2015) we observed unequal labeling, leading to far more LexA_op_ labelled cells than GAL4 labelled cells, resulting in far fewer RFP positive cells, compounded by the RFP signal being weaker than the GFP fluorescence. To overcome this, we generated a fly line in our lab that had nuclear-localized fluorescent proteins for better identification of polyploid nuclei and switched the colors of the lexA and Gal4 expressing cells (LexA_op_-RFP_nls_ and UAS-GFP_nls_). This greatly improved our ability to resolve double labeled cells (Fiugre 5—figure supplement 1). We performed our quantification in 14 day animals because older brains frequently have aggregates of fluorescent proteins that show up as puncta in images (see Author response image 2). We think this is because RFP accumulation over time eventually becomes toxic to cells. We have performed flow cytometry on 21d Coin-FLP labeled optic lobes and indeed we see an increase of polyploid, double-labeled CoinFLP cells from 4.4% (14d) to 7% (21d). We did not include this data since we only had one replicate and we have not performed additional quantifications at later ages due to the lab Covid-19 shut down.

**Author response image 2. respfig2:** 

14) For the data presented in Figure 4, the labeling of 300 cells/optic lobe is impressive. However, if there are ~20,000 cells/optic lobe, then the 300 cells would be only ~1-2% of the total, not the estimated 25% that you measure as >4N. Can you explain the discrepancy? (BTW, we think lineage tracing methods such as Permatwin under-report dividing cells in the adult brain).

We thank the reviewer for encouraging us to explain the calculations more thoroughly. We do agree that the number of 300 per lobe is lower than expected and we think that this is, in part, because we were very conservative in our quantification and identification of double labelled cells in images. However, the numbers actually are roughly in line with what we observe via imaging and flow cytometry when the stochastic nature and biases of CoinFLP labeling are taken into account.

Here is a sample calculation to explain: If we assume 100% of cells “flip” to label with CoinFLP (which we know based on FACS and imaging we are not approaching 100%) the flipping of LexA (red) : GAL4 (green) ratio is biased 4:1 (Bosch et al., 2015). Thus if flipping is complete, we would expect 80% of diploid cells to label red and ~20% to label green. If (for simplicity) we assume that all the polyploid cells are tetraploid, and that 20% of the cells in the OL (20% of 30,000 = 6000 cells) are tetraploid, 4% of all polyploid cells will label green/green (probability of green is 0.2 therefore 0.2* 0.2=0.04*100), 64% red/red (probability of red is 0.8 thus, 0.8*0.8=0.64*100) and 32% will label green/red or red/green and appear yellow or “double-labelled”. The expectation would be then, that 1,920 cells will label yellow per optic lobe under complete 100% flipping conditions. However, as mentioned above, with our heat-shock protocol we do not label 100% of cells. The amount of flipping is variable from animal to animal and we estimate that in our samples with the lowest flipping we flip about ⅓ of cells and with our strongest flipping we label about ¾ cells. If we label ~50% of cells, we would expect about 900 yellow cells per optic lobe, which is much closer to what we observe for CoinFLP by FACS and in our imaging (Fiugre 5—figure supplement 1 and 2). We have tabulated some of these numbers comparing CoinFLP quantification by imaging and by FACS in Author response table 1 to clarify.

**Table resptable1:** 

	Expected theoretical outcome (100% flipping)	Expected theoretical outcome (70% flipping)	Expected theoretical outcome (50% flipping)	Actual outcome from imaging and manual counting	Actual outcome measured by flow cytometry (~70% labelled)
Diploid cells flipping red	80%	56%	40%	ND	54.3%
Diploid cells flipping green	20%	14%	10%	ND	11.9%
Ratio of red:green	4:1	4:1	4:1	ND	4.6:1
% double labelled nuclei per OL	6.4%	4.48%	3.2%	1-2%	4.4%
“Yellow” cells per OL	1,920	1,344	960	~317	1,320

15) For Figure 6, it would be interesting to pulse the flies with paraquat, then chase a few days without paraquat before assessing ploidy. What if the polyploid cells are en route to dying? This seems unlikely for UV irradiated flies, b/c the irradiation was done 5 days prior to dissection. However, in Figure 5D, the fact that the control and paraquat-treated lines cross at 21 days seems to support the idea that some of the polyploid cells in the paraquat-treated animals die and/or divide. Please comment.

This would be an interesting experiment to perform because we think that the animals cultured on low dose of PQ for an extended period of time may be exhibiting adaptations to chronic oxidative stress. In Figure 6D, we do observe a slightly lower % of polyploidy at 21d than at 14d, but this is not statistically significantly different from either of those two timepoints. We interpret the results of this experiment as flies treated with PQ reaching “peak” polyploidy levels earlier than their untreated counterparts (at 7d instead of 21 or 28). We also see (Figure 6—figure supplement 1) that there is no significant difference in % of cell death between control and PQ treated animals. However, in all time points (21d shown below), most of the dying cells are diploid.

16) In Figure 6—figure supplement 1. It appears that pH2AV levels are elevated in all neurons in the paraquat treated brains. However, only a fraction of these die. Why might this be?

This is in interesting observation and there are two parts to address. The first is specific to our paraquat treatment. The dose is intentionally low to mimic chronic oxidative stress with age and this might be inducing a low “below threshold” level of DNA damage in neurons. The second part to address is that this is widely observed with DNA damage – where damage occurs in many cells, but only some cells of a tissue die while others can be protected from death, despite sustaining high levels of DNA damage. Work from other labs has shown that not all the cells exposed to DNA damage undergo cell death and that various factors such as cell type, cell signaling etc. can influence this (Jaklevic and Su, 2004; Verghese and Su, 2017). Finally, recent work from the Abrams lab (DOI: 10.1091/mbc.E18-12-0791) has shown that high levels of DNA damage to the postmitotic adult fly brain does not induce cell death with the canonical kinetics established in embryos or wings. Our data also agrees with this and instead of death at 1-8h post damage (which they examine) we observe very delayed kinetics (16-24h post damage, see new data in Figure 6) with death resolving after 24h, despite high levels of DNA damage persisting. The Abrams work attributes this lack of cell death to noncanonical functions of p53 in the adult brain (which undoubtedly is true), but here we show there actually is an acute death response in the adult brain, at time later than what they examined. We believe this response is likely p53independent. There is a whole area of work to be done here to resolve how postmitotic tissues signal and cope with DNA damage, which we hope to pursue in the future.